# Provably Robust Score-Based Diffusion Posterior Sampling for Plug-and-Play Image Reconstruction

**Xingyu Xu**[*]
Carnegie Mellon University

**Yuejie Chi**[†]
Carnegie Mellon University

## Abstract

In a great number of applications, the goal is to infer an unknown image from a small number of noisy measurements collected from a known and possibly nonlinear forward model describing certain sensing or imaging modality, which is often ill-posed. Score-based diffusion models, thanks to their impressive empirical success, have emerged as an appealing candidate of an expressive prior in image reconstruction. In order to accommodate diverse tasks at once, it is of great interest to develop efficient, consistent and robust algorithms that incorporate *unconditional* score functions of an image prior distribution in conjunction with flexible choices of forward models. This work develops an algorithmic framework for employing score-based diffusion models as an expressive data prior in nonlinear inverse problems with general forward models. Motivated by the plug-and-play framework in the imaging community, we introduce a diffusion plug-and-play method (DPnP) that alternatively calls two samplers, a proximal consistency sampler based solely on the likelihood function of the forward model, and a denoising diffusion sampler based solely on the score functions of the image prior. The key insight is that denoising under white Gaussian noise can be solved *rigorously* via both stochastic (i.e., DDPM-type) and deterministic (i.e., DDIM-type) samplers using the same set of score functions trained for generation. We establish both asymptotic and non-asymptotic performance guarantees of DPnP, and provide numerical experiments to illustrate its promise in various tasks. To the best of our knowledge, DPnP is the first provably-robust posterior sampling method for nonlinear inverse problems using unconditional diffusion priors.

## 1 Introduction

In a great number of sensing and imaging applications, the paramount goal is to infer an unknown image $x^\star \in \mathbb{R}^d$ from a collection of measurements $y \in \mathbb{R}^m$ that are possibly noisy, incomplete, and even nonlinear. Examples include restoration tasks such as inpainting, super-resolution, denoising, as well as imaging tasks such as magnetic resonance imaging [LDP07], optical imaging [SEC+15], microscopy imaging [HSMC17], radar and sonar imaging [PEPC10], and many more.

Due to sensing and resource constraints, the problem of image reconstruction is often ill-posed, where the desired resolution of the unknown image overwhelms the set of available observations. Consequently, this necessitates the need of incorporating prior information regarding the unknown image to assist the reconstruction process. Over the years, numerous types of prior information have been considered and adopted, from hand-crafted priors such as subspace or sparsity constraints [Don06, CR12], to data-driven ones prescribed in the form of neural networks [UVL18, BJPD17]. These priors can be regarded as some sort of generative models for the unknown image, which postulate the high-dimensional image admits certain parsimonious representation in a low-dimensional data manifold. It is desirable that the generative models are sufficiently expressive to capture the

---

[*]`xingyuxu@andrew.cmu.edu`

[†]`yuejiec@andrew.cmu.edu`

38th Conference on Neural Information Processing Systems (NeurIPS 2024).

| Super-resolution (linear) | Phase retrieval (nonlinear) | Quantized sensing (nonlinear) |
|---|---|---|

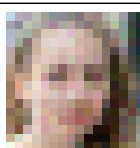 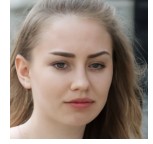 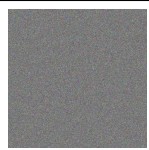 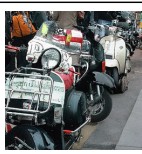 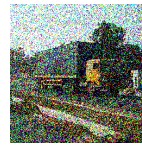 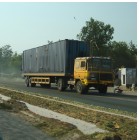

Figure 1: Solving linear and nonlinear inverse problems with Diffusion Plug-and-Play (DPnP).

diversity and structure of the image class of interest, yet nonetheless, still lead to image reconstruction problems that are computationally tractable.

**Score-based diffusion models as an image prior.** Recent years have seen tremendous progress on generative artificial intelligence (AI), where it is possible to generate new data samples — such as images, audio, text — at unprecedented resolution and scale from a target distribution given training data. Diffusion models, originally proposed by [SDWMG15], are among one of the most successful frameworks, underneath popular content generators such as DALL·E [RDN+22], Stable Diffusion [RBL+22], Imagen [SCS+22], and many others. Roughly speaking, score-based diffusion models convert noise into samples that resemble those from a target data distribution, by forming the reverse Markov diffusion process only using the score functions of the data contaminated at various noise levels [SE19, HJA20, SSDK+21, SME20]. In particular, [SME20] developed a unified framework to interpret score-based diffusion models as reversing certain Stochastic Differential Equations (SDE) using either SDE or probability flow Ordinary Differential Equations (ODE), leading to stochastic (i.e., DDPM-type) and deterministic (i.e., DDIM-type) samplers, respectively. While the DDIM-type sampler is more amenable to acceleration, the DDPM-type sampler tends to generate images of higher quality and diversity when running for a large number of steps [SME20].

Thanks to the expressive power of score-based diffusion models in generating complex and fine-grained images, they have emerged as a plausible candidate of an expressive prior in image reconstruction [SSXE21, CKM+23, FSR+23] via the lens of *Bayesian posterior sampling*. To accommodate diverse applications with various image characteristics and imaging modalities, it is desirable to develop *plug-and-play* methods that do not require training from scratch or end-to-end training for every new imaging task. Nonetheless, despite a flurry of recent efforts, existing algorithms either are computationally expensive [WTN+23, CICM23], inconsistent [CKM+23, KEES22, MSKV24], or confined to linear inverse problems [CICM23, DS24]. Therefore, a natural question arises:

*Can we develop a practical, consistent and robust algorithm that incorporates score-based diffusion models as an image prior with general (possibly nonlinear) forward models?*

## 1.1 Our contribution

This paper provides an affirmative answer to this question, by developing an algorithmic framework to sample from the posterior distribution of images, where score-based diffusion models are employed as an expressive image prior in nonlinear inverse problems with general forward models. Specifically, our contributions are as follows.

- *Diffusion plug-and-play for posterior sampling.* Motivated by the plug-and-play [VBW13] framework in the imaging community, we introduce a diffusion plug-and-play method (DPnP) that alternatively calls two samplers, a *proximal consistency sampler* that aims to generate samples that are more consistent with the measurements, and a *denoising diffusion sampler* that focuses on sampling from the posterior distribution of an easier problem — image denoising under white Gaussian noise — to enforce the prior constraint. Our method is *modular*, in the sense that the *proximal consistency sampler* is solely based on the likelihood function of the forward model, and the denoising diffusion sampler is based solely on the score functions of the image prior.

- *Posterior sampling for image denoising.* While the proximal consistency sampler can be borrowed somewhat straightforwardly from existing literature such as the Metropolis-adjusted Langevin algorithm [RR98], the denoising diffusion sampler, on the other hand, has not been addressed in the literature to the best of our knowledge. Our key insight is that this can be solved via both stochastic (i.e., DDPM-type) or deterministic (i.e., DDIM-type) samplers by carefully choosing the forward SDEs and discretizing the resulting reversal SDE or ODE using the exponential integrator [ZC22]. Importantly, the denoising diffusion samplers use the same set of unconditional score functions for generation, making it readily implementable without additional training.

- *Theoretical guarantees.* We establish both asymptotic and non-asymptotic performance guarantees of the proposed DPnP method. Asymptotically, we verify the correctness of our method by proving that DPnP converges to the conditional distribution of $x^\star$ given measurements $y$, assuming exact unconditional score estimates of the image prior. We next establish a non-asymptotic convergence theory of DPnP, where its performance degenerates gracefully with respect to the errors of the samplers, due to, e.g., score estimation errors and limited sampling steps. To the best of our knowledge, this provides the *first provably-robust* method for *nonlinear* inverse problems using unconditional score-based diffusion priors.

We further provide numerical experiments to illustrate its promise in solving both linear and nonlinear image reconstruction tasks, such as super-resolution, phase retrieval, and quantized sensing. Due to its plug-and-play nature, we expect it to be of broad interest to a wide variety of inverse problems.

**Related works.** Given its interdisciplinary nature, our work sits at the intersection of generative modeling, computational imaging, optimization and sampling. Due to space limits, we postpone the discussion of related works to Appendix A.

**Notation.** Let $p_x$ denote the probability distribution of $x$, and $p_x(\cdot|y)$ denotes the conditional distribution of $x$ given $y$. We use $X \stackrel{\text{(d)}}{=} Y$ to denote random variables $X$ and $Y$ are equivalent in distribution. The matrix $I_d$ denotes an identity matrix of dimension $d$. For two probability distributions with density $p(x)$ and $q(x)$, the total variation distance between them is $\mathsf{TV}(p, q) :=$ $\int |p(x) - q(x)| \mathrm{d}x$. The $\chi^2$-divergence of $p$ to $q$ is $\chi^2(p \,\|\, q) := \int \frac{(p(x) - q(x))^2}{q(x)} \mathrm{d}x$.

## 2 Score-based generative models

In this section, we set up the preliminary on diffusion-based generative models, which we will be relying upon to develop our algorithm. The key components consist of a *forward* process, which diffuses the data distribution $p^\star$ to the standard normal distribution by gradually injecting noise into the samples, and a *backward* process, which reverses the forward process so that it can transform the standard normal distribution to the data distribution $p^\star$. To facilitate understanding, it will be convenient to formulate these processes in continuous time. For discrete-time formulation and implementation, please refer to Appendix E.

### 2.1 The forward process and score functions

The continuous-time forward diffusion follows the Ornstein-Uhlenbeck (OU) process, defined by the Stochastic Differential Equation (SDE) [SSDK$^+$21]:

$$\mathrm{d}X_\tau = -X_\tau \mathrm{d}\tau + \sqrt{2}\,\mathrm{d}B_\tau, \quad \tau \geq 0, \quad X_0 \sim p^\star, \tag{1}$$

where $(B_\tau)_{\tau \geq 0}$ is the standard $d$-dimensional Brownian motion. It can be shown that [Doo42, Eva12] the marginal distribution of $X_\tau$ for $\tau \geq 0$ is

$$X_\tau \stackrel{\text{(d)}}{=} \mathrm{e}^{-\tau} X_0 + \sqrt{1 - \mathrm{e}^{-2\tau}}\,\varepsilon, \quad X_0 \sim p^\star, \ \varepsilon \sim \mathcal{N}(0, I_d). \tag{2}$$

It is then clear that the limiting distribution $X_\infty \sim \mathcal{N}(0, I_d)$ as $\tau \to \infty$, i.e., the OU process diffuses $X_0 \sim p^\star$ to the standard normal distribution. The score function of $X_\tau$ is defined by

$$s(\tau, x) = \nabla \log p_{X_\tau}(x). \tag{3}$$

An enlightening property [Vin11] of the score function is that it can be interpreted as the minimum mean-squared error (MMSE) estimate of $\varepsilon_t$ given $x_t = x$, fueled by Tweedie's formula:

$$s(\tau, x) = -\frac{1}{\sqrt{1 - \mathrm{e}^{-2\tau}}} \underbrace{\mathbb{E}_{X_0 \sim p^\star, \varepsilon \sim \mathcal{N}(0, I_d)}\big(\varepsilon \,|\, \mathrm{e}^{-\tau} X_0 + \sqrt{1 - \mathrm{e}^{-2\tau}}\varepsilon = x\big)}_{=:\varepsilon(\tau, x)} \tag{4}$$

Consequently, this makes it possible to estimate the score functions via learning to denoise [Hyv05], by estimating the denoising function $\varepsilon(\tau, \cdot)$, as typically done in practice [HJA20].

### 2.2 The reverse process and sampling

To enable sampling, one needs to "reverse" the forward diffusion process. Fortunately, it is possible to leverage classical theory [And82, AGS05] to reverse the SDE, and apply discretization to the time-reversal processes to collect samples. We shall describe two popular approaches below, corresponding to stochastic (i.e., DDPM-type) and deterministic (i.e., DDIM-type) samplers respectively following primarily the framework set forth in [SSDK$^+$21].

**Time-reversed SDEs and probability flow ODEs.** Let us begin with the more general theory of *reversing* SDEs, which will be useful in future sections. Consider a SDE given by

$$\mathrm{d}M_\tau = \alpha M_\tau \mathrm{d}\tau + \sqrt{\beta}\mathrm{d}B_\tau, \quad \tau \geq 0, \quad M_0 \sim p_{M_0}, \tag{5}$$

where $\alpha \in \mathbb{R}$ and $\beta > 0$ are constants. For any positive time $\tau_\infty > 0$, define the reversed time parameter

$$\tau^{\mathsf{rev}} := \tau^{\mathsf{rev}}(\tau) = \tau_\infty - \tau. \tag{6}$$

We are now ready to describe the time-reversed processes.

1) The *time-reversed SDE* of (5) on the time interval $[0, \tau_\infty]$ is defined as

$$\mathrm{d}M_{\tau^{\mathsf{rev}}}^{\mathsf{rev}} = (-\alpha M_{\tau^{\mathsf{rev}}}^{\mathsf{rev}} + \beta \nabla \log p_{M_{\tau^{\mathsf{rev}}}}(M_{\tau^{\mathsf{rev}}}^{\mathsf{rev}})) \mathrm{d}\tau + \sqrt{\beta}\mathrm{d}\tilde{B}_\tau, \ \tau \in [0, \tau_\infty], \ M_{\tau_\infty}^{\mathsf{rev}} \sim p_{M_{\tau_\infty}}, \tag{7}$$

where $\tilde{B}$ is an independent copy of $B$, i.e., another Brownian motion. It is a classical result [And82] that the reversed process $M^{\mathsf{rev}}$ shares the same path distribution as $M$, i.e., $(M_\tau^{\mathsf{rev}})_{\tau \in [0, \tau_\infty]} \overset{(\mathrm{d})}{=} (M_\tau)_{\tau \in [0, \tau_\infty]}$. In other words, the joint distribution of $(M_{\tau_1}^{\mathsf{rev}}, M_{\tau_2}^{\mathsf{rev}}, \cdots, M_{\tau_k}^{\mathsf{rev}})$ for any $0 \leq \tau_1 \leq \tau_2 \leq \cdots \leq \tau_k \leq \tau_\infty$, for any integer $k \geq 1$, coincides with that of $(M_{\tau_1}, M_{\tau_2}, \cdots, M_{\tau_k})$.

2) In place of the reversed SDE in (7), it is possible to consider the following probability flow ODE [AGS05, SSDK+21]:

$$\mathrm{d}M_{\tau^{\mathsf{rev}}}^{\mathsf{rev}} = \left(-\alpha M_{\tau^{\mathsf{rev}}}^{\mathsf{rev}} + \frac{\beta}{2} \nabla \log p_{M_{\tau^{\mathsf{rev}}}}(\tau^{\mathsf{rev}}, M_{\tau^{\mathsf{rev}}}^{\mathsf{rev}})\right) \mathrm{d}\tau, \ \tau \in [0, \tau_\infty], \ M_{\tau_\infty}^{\mathsf{rev}} \sim p_{M_{\tau_\infty}}. \tag{8}$$

The reversed ODE satisfies a slightly weaker guarantee than that of the reversed SDE, which nevertheless suffices for most practical purposes [SSDK+21]: $M_\tau^{\mathsf{rev}} \overset{(\mathrm{d})}{=} M_\tau, \quad \tau \in [0, \tau_\infty]$. Note that the reversed ODE only guarantees identical marginal distribution for each $M_\tau^{\mathsf{rev}}$, whereas the reversed SDE guarantees identical joint distribution.

Specializing the above to the OU process (1) with proper discretization then leads to popular samplers used for generation, as follows.

**DDPM-type stochastic samplers.** Specializing the time-reversed SDE (7) to the OU process gives

$$\mathrm{d}X_{\tau^{\mathsf{rev}}}^{\mathsf{rev}} = \left(X_{\tau^{\mathsf{rev}}}^{\mathsf{rev}} + 2s(\tau^{\mathsf{rev}}, X_{\tau^{\mathsf{rev}}}^{\mathsf{rev}})\right)\mathrm{d}\tau + \sqrt{2}\mathrm{d}\tilde{B}_\tau, \quad \tau \in [0, \tau_\infty], \quad X_{\tau_\infty}^{\mathsf{rev}} \sim p_{X_{\tau_\infty}}.$$

As $\tau_\infty \to \infty$, it can be seen from (2) that $p_{X_{\tau_\infty}}$ converges to $\mathcal{N}(0, I_d)$. Thus the solution of the above SDE can be approximated by initializing $X_{\tau_\infty}^{\mathsf{rev}} \sim \mathcal{N}(0, I_d)$ instead. The DDPM sampler [HJA20] can be viewed as a discretization of this SDE [SSDK+21].

**DDIM-type deterministic samplers.** On the other hand, the probability flow ODE (8) for the OU process reads as

$$\mathrm{d}X_{\tau^{\mathsf{rev}}}^{\mathsf{rev}} = \left(X_{\tau^{\mathsf{rev}}}^{\mathsf{rev}} + s(\tau^{\mathsf{rev}}, X_{\tau^{\mathsf{rev}}}^{\mathsf{rev}})\right)\mathrm{d}\tau, \quad \tau \in [0, \tau_\infty], \quad X_{\tau_\infty}^{\mathsf{rev}} \sim p_{X_{\tau_\infty}}. \tag{9}$$

Again, as $\tau_\infty \to \infty$, one may approximate the initialization with $X_{\tau_\infty}^{\mathsf{rev}} \sim \mathcal{N}(0, I_d)$. It is known that the popular DDIM sampler [SSDK+21, SME20] is a discretization of this ODE [ZC22]. The ODE-based deterministic samplers allow more aggressive choice of discretization schedules, as well as fast ODE solvers [LZB+22], enabling significantly accelerated sampling process compared to the SDE-based stochastic samplers.

## 3 Posterior sampling via diffusion plug-and-play

We are interested in solving (possibly nonlinear) inverse problems, where the aim is to infer an unknown image $x^\star \in \mathbb{R}^d$ from its measurements $y \in \mathbb{R}^m$,

$$y = \mathcal{A}(x^\star) + \xi,$$

where $\mathcal{A} : \mathbb{R}^d \to \mathbb{R}^m$ is the measurement operator underneath the forward model, and $\xi$ denotes measurement noise. We focus on the Bayesian setting where the prior information of $x^\star$ is provided in the form of some prior distribution $p^\star(\cdot)$, i.e.,

$$x^\star \sim p^\star(x), \tag{10}$$

The *posterior distribution* given measurements $y$ is defined as

$$p^\star(x|y) \propto p^\star(x)\, p(y|x^\star = x) = p^\star(x)\, \mathrm{e}^{\mathcal{L}(x;y)}. \qquad (11)$$

Here, $\mathcal{L}(\cdot\,; y)$ is the log-likelihood function of the measurements. Notwithstanding, our framework allows flexible choices of the forward model and the noise distributions. In addition, while this formulation is derived from probabilistic interpretations, it also subsumes the "reward-guided" or "loss-guided" setting [SZY$^+$23], where $\mathcal{L}$ can be viewed as a reward function or a negative loss function, both of which characterize preference over structural properties of $x^\star$.

**Assumption on the forward model.** Throughout the paper, for simplicity, we make the following mild assumption on $\mathcal{L}$, which is applicable to many applications of interest.

**Assumption 1.** *We assume $\mathcal{L}(\cdot\,; y)$ is differentiable almost everywhere, and $\sup_{x\in\mathbb{R}^d} \mathcal{L}(x;y) < \infty$.*

**Goal.** Our goal is to sample $\widehat{x}$ from the posterior distribution $\widehat{x} \sim p^\star(\cdot \,|\, y)$ given estimates $\widehat{s}(\tau, x)$ (resp. $\widehat{\varepsilon}(\tau, x)$) of the *unconditional* score functions $s(\tau, x)$ (resp. the noise function $\varepsilon(\tau, x)$) in (3), assuming knowledge of the likelihood function $\mathcal{L}(\cdot\,; y)$.

### 3.1 Key ingredient: score-based denoising posterior sampling

We begin with an inspection on one of the most fundamental inverse problems: denoising under white Gaussian noise. As shall be elucidated shortly, the denoising diffusion samplers turn out to be an important building block in our algorithm for general inverse problems.

**Image denoising under white Gaussian noise.** Suppose that we have access to a noisy version of $x^\star \sim p^\star$ contaminated by white Gaussian noise, given by

$$x_{\mathsf{noisy}} = x^\star + \xi, \quad \xi \sim \mathcal{N}(0, \eta^2 I_d), \qquad (12)$$

where $\eta > 0$ is the noise intensity *assumed to be known*. Our goal is to sample from $p^\star(\cdot \,|\, x_{\mathsf{noisy}})$ given the score estimates $\widehat{s}_t(x)$ (resp. the noise estimates $\widehat{\varepsilon}_t(x)$). We will develop our score-based denoising posterior sampler, termed DDS, with two variants, DDS-DDPM and DDS-DDIM, which can be viewed as analogues of the well-known DDPM and DDIM samplers in unconditional score-based sampling respectively. Before proceeding, it is worth highlighting that the two variants will be derived from different forward diffusion processes, since we observe the resulting variants empirically lead to more competitive performance.

**A stochastic DDPM-type sampler via heat flow.** We begin with a stochastic DDPM-type sampler for denoising, termed DDS-DDPM. We divide our development into the following steps.

1) *Step 1: introducing the heat flow.* Let us introduce a *heat flow* with initial distribution $p^\star$, defined by the following SDE:

$$\mathrm{d}Y_\tau = \mathrm{d}B_\tau, \quad \tau \geq 0, \quad Y_0 \sim p^\star, \qquad (13)$$

where $(B_\tau)_{\tau\geq 0}$ is the standard $d$-dimensional Brownian motion. The solution of (13) is simply

$$Y_\tau = Y_0 + B_\tau, \quad \tau \geq 0. \qquad (14)$$

Since $B_\tau \sim \mathcal{N}(0, \tau I_d)$, it readily follows that $B_{\eta^2} \stackrel{\mathrm{(d)}}{=} \xi$, which together with $Y_0 \sim p^\star$ yield the important observation that $x_{\mathsf{noisy}} = x^\star + \xi$ can be viewed as an endpoint of the heat flow, in the sense that $x_{\mathsf{noisy}} = x^\star + \xi \stackrel{\mathrm{(d)}}{=} Y_{\eta^2}$.

2) *Step 2: reversing the heat flow.* Following similar reasonings in Section 2, the next step boils down to reverse the heat flow (13). The time-reversal of the heat flow SDE (13) is (cf. (7)) given by

$$\mathrm{d}Y^{\mathsf{rev}}_{\eta^2-\tau} = \nabla \log p_{Y_{\eta^2-\tau}}(Y^{\mathsf{rev}}_{\eta^2-\tau})\mathrm{d}\tau + \mathrm{d}\tilde{B}_\tau, \quad \tau \in [0, \eta^2], \quad Y^{\mathsf{rev}}_{\eta^2} \sim p_{Y_{\eta^2}}, \qquad (15)$$

where $(\tilde{B}_\tau)_{\tau\geq 0}$ is an independent copy of $(B_\tau)_{\tau\geq 0}$. As introduced earlier, the virtue of the time-reversed SDE (15) is that it produces a process $Y^{\mathsf{rev}}_\tau$ with the same *path* distribution as $Y_\tau$, i.e., $(Y^{\mathsf{rev}}_\tau)_{\tau\in[0,\eta^2]} \stackrel{\mathrm{(d)}}{=} (Y_\tau)_{\tau\in[0,\eta^2]}$. In particular, the joint distribution of $(Y^{\mathsf{rev}}_0, Y^{\mathsf{rev}}_{\eta^2})$ is the same as that of $(Y_0, Y_{\eta^2}) \stackrel{\mathrm{(d)}}{=} (x^\star, x_{\mathsf{noisy}})$. This implies that the conditional distribution $p^\star(\cdot \,|\, x_{\mathsf{noisy}})$ is the same as $p_{Y^{\mathsf{rev}}_0}(\cdot \,|\, Y^{\mathsf{rev}}_{\eta^2} = x_{\mathsf{noisy}})$. Surprisingly, the latter admits a simple interpretation: $p_{Y^{\mathsf{rev}}_0}(\cdot \,|\, Y^{\mathsf{rev}}_{\eta^2} = x_{\mathsf{noisy}})$ is the distribution of $Y^{\mathsf{rev}}_0$ when we initialize (15) with $Y^{\mathsf{rev}}_{\eta^2} = x_{\mathsf{noisy}}$! Therefore, sampling the posterior $p^\star(\cdot \,|\, x_{\mathsf{noisy}})$ amounts to solving the following simple SDE:

$$\mathrm{d}Y^{\mathsf{rev}}_{\eta^2-\tau} = \nabla \log p_{Y_{\eta^2-\tau}}(Y^{\mathsf{rev}}_{\eta^2-\tau})\mathrm{d}\tau + \mathrm{d}\tilde{B}_\tau, \quad \tau \in [0, \eta^2], \quad Y^{\mathsf{rev}}_{\eta^2} = x_{\mathsf{noisy}}. \qquad (16)$$

3) *Step 3: connecting the score functions.* It is now immediate to arrive at our proposed stochastic sampler DDS-DDPM by discretization of this SDE (16), which requires knowledge of the score functions $\nabla \log p_{Y_\tau}(\cdot)$. A key observation is that they can in fact be computed from the score function $s(\tau, x)$ (cf. (3)), due to the following lemma, whose proof is provided in Appendix D.1.

**Lemma 1** (Score function of $Y_\tau$). *For $\tau \geq 0$, we have*

$$\nabla \log p_{Y_\tau}(x) = \frac{1}{\sqrt{1+\tau}} s\left(\frac{1}{2}\log(1+\tau), \frac{x}{\sqrt{1+\tau}}\right).$$

The resulting sampler, DDS-DDPM, is summarized in Algorithm 2 (deferred in the appendix) using a discretization procedure with an exponential integrator [ZC22].

**A deterministic DDIM-type sampler via OU process.** We next develop a deterministic DDIM-type sampler for denoising, termed DDS-DDIM.

1) *Step 1: introducing a posterior-initialized OU process.* To sample from the posterior distribution $p^\star(\cdot|x_{\mathsf{noisy}})$, we first introduce a random variable $w$ which has (unconditional) distribution

$$p_w(x) := p^\star(x^\star = x \mid x^\star + \xi = x_{\mathsf{noisy}}), \tag{17}$$

in the same form of the desired posterior distribution $p^\star(\cdot|x_{\mathsf{noisy}})$. Here, since the noisy observation $x_{\mathsf{noisy}}$ is given, we regard it as fixed. We then further introduce $z = w - x_{\mathsf{noisy}}$, which is a "centered" version of $w$, whose distribution is

$$p_z(x) := p_w(x + x_{\mathsf{noisy}}) = p^\star(x^\star = x + x_{\mathsf{noisy}} \mid x^\star + \xi = x_{\mathsf{noisy}}).$$

The OU process with initial distribution $p_z$ is defined by the SDE:

$$\mathrm{d}Z_\tau = -Z_\tau \mathrm{d}\tau + \mathrm{d}B_\tau, \quad \tau \geq 0, \quad Z_0 \sim p_z, \tag{18}$$

where $B_\tau$ is the standard $d$-dimensional Brownian motion. As in (2), the marginal distribution of $Z_\tau$ is given by

$$Z_\tau \overset{(\mathrm{d})}{=} \mathrm{e}^{-\tau}Z_0 + \sqrt{1 - \mathrm{e}^{-2\tau}}\varepsilon, \quad Z_0 \sim p_z, \ \varepsilon \sim \mathcal{N}(0, I_d), \quad \tau \geq 0. \tag{19}$$

2) *Step 2: reversing the OU process.* Following similar reasonings in Section 2, reversing the OU process (18) will enable us to generate samples $z \sim p_z$. Then we can set $w = z + x_{\mathsf{noisy}}$, which, by definition, has distribution $p_w$ defined in (17), and is a sample from the desired posterior distribution $p^\star(\cdot|x_{\mathsf{noisy}})$. We are thus led to solve the time-reversed probability flow ODE (cf. (8)) of (18), given by

$$\mathrm{d}Z^{\mathsf{rev}}_{\tau^{\mathsf{rev}}} = \left(Z^{\mathsf{rev}}_{\tau^{\mathsf{rev}}} + \nabla \log p_{Z_{\tau^{\mathsf{rev}}}}(\tau^{\mathsf{rev}}, Z^{\mathsf{rev}}_{\tau^{\mathsf{rev}}})\right)\mathrm{d}\tau, \quad \tau \in [0, \tau_\infty], \quad Z^{\mathsf{rev}}_{\tau_\infty} \sim \mathcal{N}(0, I_d), \quad \tau^{\mathsf{rev}} = \tau_\infty - \tau. \tag{20}$$

3) *Step 3: connecting the score functions.* We are now one step away from our proposed deterministic sampler DDS-DDIM, which is derived by discretization of the ODE (20). We need to know the score functions $\nabla \log p_{Z_\tau}(\cdot)$, which again can be computed from the score function $s(\tau, x)$ (cf. (3)), as documented by the following lemma, whose proof is provided in Appendix D.2.

**Lemma 2** (Score function of $Z_\tau$). *For $\tau \geq 0$, we have*

$$\nabla \log p_{Z_\tau}(x) = -\frac{\mathrm{e}^{2\tau}x}{\eta^2 + \mathrm{e}^{2\tau} - 1} + \frac{\mathrm{e}^{\tau - \tilde{\tau}}\eta^2}{\eta^2 + \mathrm{e}^{2\tau} - 1} s\left(\tilde{\tau}, \mathrm{e}^{-\tilde{\tau}}x_{\mathsf{noisy}} + \frac{\mathrm{e}^{\tau - \tilde{\tau}}\eta^2 x}{\eta^2 + \mathrm{e}^{2\tau} - 1}\right), \tag{21}$$

*where*

$$\tilde{\tau} := \tilde{\tau}(\tau) = \frac{1}{2}\log\left(\frac{\eta^2(\mathrm{e}^{2\tau} - 1)}{\eta^2 + \mathrm{e}^{2\tau} - 1} + 1\right). \tag{22}$$

After plugging this into (20) and solving the ODE for $Z^{\mathsf{rev}}_\tau$, we see that $Z^{\mathsf{rev}}_0 + x_{\mathsf{noisy}}$ is the desired sample from the posterior distribution $p^\star(\cdot|x_{\mathsf{noisy}})$, as argued before. Numerically, the ODE (20) is solved by discretization with an exponential integrator [ZC22], resulting in the sampler DDS-DDIM as summarized in Algorithm 3 (deferred in the appendix).

---

**Algorithm 1** Diffusion Plug-and-Play ($\mathsf{DPnP}$)

---

**Input**: Measurements $y \in \mathbb{R}^m$, log-likelihood function $\mathcal{L}(\cdot\,;y)$ of the forward model, score estimates $\widehat{s}$, annealing schedule $(\eta_k)_{0 \leq k \leq K}$.

**Initialization**: Sample $\widehat{x}_0 \sim \mathcal{N}(0, \frac{\eta_0}{4} \overline{I}_d)$

**Alternating sampling**: **for** $k = 0, 1, 2, \ldots, K-1$ **do**

(1) *Proximal consistency sampler:* Sample $\widehat{x}_{k+\frac{1}{2}} \propto \exp\left(\mathcal{L}(\cdot\,;y) - \frac{1}{2\eta_k^2}\|\cdot - \widehat{x}_k\|^2\right)$ using subroutine $\mathsf{PCS}(\widehat{x}_k, y, \mathcal{L}, \eta_k)$ (Alg. 4).

(2) *Denoising diffusion sampler:* Sample $\widehat{x}_{k+1} \sim \exp\left(\log p^\star(x) - \frac{1}{2\eta_k^2}\|x - \widehat{x}_{k+\frac{1}{2}}\|^2\right)$ using subroutine $\mathsf{DDS\text{-}DDPM}(\widehat{x}_{k+\frac{1}{2}}, \widehat{s}, \eta_k)$ (Alg. 2) or $\mathsf{DDS\text{-}DDIM}(\widehat{x}_{k+\frac{1}{2}}, \widehat{s}, \eta_k)$ (Alg. 3).

**Output**: $\widehat{x}_K$.

---

### 3.2 Our algorithm: diffusion plug-and-play

Now we turn to the general setting where the measurement operator $\mathcal{A}$ is arbitrary. From the factorization of posterior distribution in (11), one intuitively understands that a posterior sampler must obey two constraints simultaneously: (i) the *data prior constraint*, corresponding to the first factor $p^\star(x)$, which imposes that the posterior sampler should be less likely to sample at those points where $p^\star(x)$ is small; (ii) the *measurement consistency constraint*, corresponding to the second factor $e^{\mathcal{L}(x;y)}$, which imposes that $\mathcal{A}(x) \approx y$.

**Diffusion plug-and-play ($\mathsf{DPnP}$).** We will apply the idea of alternatively enforcing these two constraints from a sampling perspective in the same spirit of [VDC19, LST21, BB23]. Our algorithm, dubbed diffusion plug-and-play ($\mathsf{DPnP}$), alternates between two samplers, the denoising diffusion sampler ($\mathsf{DDS}$) and the proximal consistency sampler ($\mathsf{PCS}$), which can be viewed as the substitutes for the proximal operator and the gradient step respectively. Given the iterate $\widehat{x}_k$ and the *annealing* parameter $\eta_k$ at the $k$-th iteration, $\mathsf{DPnP}$ proceeds with the following two steps:

(i) *Proximal consistency sampler to enforce the measurement consistency constraint.* $\mathsf{DPnP}$ draws a sample $\widehat{x}_{k+\frac{1}{2}}$ from the distribution proportional to $\exp\left(\mathcal{L}(x\,;y) - \frac{1}{2\eta_k^2}\|x - \widehat{x}_k\|^2\right)$ to promote the image to be consistent with the measurements. This step, which we denote as the *proximal consistency sampler*, can be achieved by small modifications of standard algorithms such as Metropolis-Adjusted Langevin Algorithm (MALA) [RR98] given in Algorithm 4 (deferred in the appendix).

(ii) *Denoising diffusion sampler to enforce the data prior constraint.* $\mathsf{DPnP}$ next draws a sample $\widehat{x}_{k+1}$ from the distribution proportional to

$$\exp\left(-\left(-\log p^\star(x) + \frac{1}{2\eta_k^2}\|x - \widehat{x}_{k+\frac{1}{2}}\|^2\right)\right) \propto p^\star(x^\star = x \mid x^\star + \eta_k w = \widehat{x}_{k+\frac{1}{2}}) \qquad (23)$$

to promote the image to be consistent with the prior, where $w \sim \mathcal{N}(0, I_d)$. The last step, which follows from the Bayes' rule, makes it clear that this step can be precisely achieved by the denoising diffusion sampler (developed in Section 3.1) using solely the unconditional score function.

Combining both steps lead to the proposed $\mathsf{DPnP}$ method described in Algorithm 1. Some comments about the proposed $\mathsf{DPnP}$ method are in order.

- The proximal consistency sampler $\mathsf{PCS}$ can be viewed as a "soft" version of the proximal point method [Dru17]. This can be seen from a first-order approximation: the maximum likelihood of the distribution $\exp\left(\mathcal{L}(\cdot\,;y) - \frac{1}{2\eta_k^2}\|\cdot - \widehat{x}_k\|^2\right)$ is attained at the point $x' \in \mathbb{R}^d$ satisfying

$$\nabla_{x'}\mathcal{L}(x';y) - \frac{1}{\eta_k^2}(x' - \widehat{x}_k) = 0, \quad \Longrightarrow \quad x' = \widehat{x}_k + \eta_k^2 \nabla_{x'}\mathcal{L}(x';y) \approx \widehat{x}_k + \eta_k^2 \nabla_{\widehat{x}_k}\mathcal{L}(\widehat{x}_k;y),$$

Therefore, the proximal consistency sampler draws random samples "concentrated" around $x'$, which approximates the implicit proximal point update, akin to a gradient step at $\widehat{x}_k$.

- On the other end, the denoising posterior sampler $\mathsf{DDS}$ can be regarded as a "soft" version of the proximal operator. In particular, when $p^\star$ is supported on a low-dimensional manifold $\mathcal{M}$, it forces $x$ to reside in $\mathcal{M}$, like the proximal map. To see this, note that denoising posterior distribution vanishes outside $\mathcal{M}$ by (23).

- The proximal consistency sampler PCS admits a simple form when the forward model $\mathcal{A}$ is linear, i.e. $\mathcal{A}(x) = Ax$ for some matrix $A \in \mathbb{R}^{m \times d}$, and the measurement noise $\xi \sim \mathcal{N}(0, \Sigma)$ is Gaussian. In this situation, the proximal consistency sampler PCS can be implemented directly by

$$\widehat{x}_{k+\frac{1}{2}} = \mathsf{PCS}(\widehat{x}_k, y, \mathcal{L}, \eta_k) = \widetilde{x}_k + \widetilde{\Sigma}_k^{1/2} w_k, \quad w_k \sim \mathcal{N}(0, I_d),$$

where $\widetilde{x}_k = \left( A^\top \Sigma^{-1} A + \frac{1}{\eta_k^2} I_d \right)^{-1} \left( A^\top \Sigma^{-1} y + \frac{1}{\eta_k^2} \widehat{x}_k \right)$, and $\widetilde{\Sigma}_k = \left( A^\top \Sigma^{-1} A + \frac{1}{\eta_k^2} I_d \right)^{-1}$.

## 4  Theoretical analysis

In this section, we establish both asymptotic and non-asymptotic performance guarantees of DPnP.

**Asymptotic consistency.** We begin with the asymptotic consistency of DPnP in the theorem below.

**Theorem 1** (Asymptotic consistency of DPnP). *Assume the score function estimate $\widehat{s}(\tau, \cdot)$ is accurate, i.e., $\widehat{s}(\tau, x) = s(\tau, x)$, and assume the ODE/SDEs in DDS and PCS are solved exactly. Let $(\varepsilon_l)_{l \geq 0}$ be a decreasing sequence of positive numbers satisfying $\lim_{l \to \infty} \varepsilon_l = 0$, and $(k_l)_{l \geq 0}$ be an increasing sequence of integers with $k_0 = 0$. Set the annealing schedule $\eta_k = \varepsilon_l$, for $k_{l-1} \leq k < k_l$, $l = 1, 2, \cdots$ Let $\min_{l'=1,2,\cdots} |k_{l'} - k_{l'-1}| \to \infty$, the output $\widehat{x}_{k_l}$ of DPnP converges in distribution to the posterior distribution $p^\star(\cdot|y)$ for $l \to \infty$.*

In words, Theorem 1 establishes the asymptotic consistency of DPnP under fairly mild assumptions on the forward model (cf. Assumption 1): as long as the sampled distributions of DDS and PCS are exact, then running DPnP with a slowly diminishing annealing schedule of $\{\eta_k\}$ will output samples approaching the desired posterior distribution $p^\star(\cdot|y)$ when the number of iterations $l$ goes to infinity.

**Non-asymptotic error analysis.** We now step away from the idealized setting when the sampled distributions of DDS and PCS are exact. In practice, there are many sources of errors that can influence the sampled distributions of DDS and PCS, e.g., the discretization error arising from numerically solving ODE/SDE, and the score estimation error. In effect, these non-idealities will make PCS and DDS *inexact*. That is, the distribution they generate will slightly deviate from the distribution they ought to sample from. In this paper, we model such deviations by the *total variation* distance from the distribution generated by PCS (resp. DDS) to the ideal distribution proportional to $\exp(\mathcal{L}(x; y) - \frac{1}{2\eta_k^2} \|x - \widehat{x}_k\|^2)$ (resp. $p^\star(x^\star = x | x^\star + \eta_k \varepsilon = \widehat{x}_{k+\frac{1}{2}})$) uniformly over all iterations. Analyzing these errors is out of the scope of this paper, and we point the interested readers to parallel lines of works, e.g., [LWCC23, MV19, CLA+21], among many others. In our analysis, we will assume a black-box bound for the total variation errors of PCS and DDS, which can be combined with existing analyses of the respective samplers to bound the iteration complexity of DPnP.

**Theorem 2** (Non-asymptotic robustness of DPnP). *With the notation in DPnP (Algorithm 1), set $\eta_k \equiv \eta > 0$. Under Assumption 1, there exists $\lambda := \lambda(p^\star, \mathcal{L}, \eta) \in (0, 1)$, such that the following holds. Define a stationary distribution $\pi_\eta$ by $\pi_\eta(x) \propto p^\star(x) q_\eta(x)$, where $q_\eta$ is defined by*

$$q_\eta(x) := e^{\mathcal{L}(\cdot; y)} * p_{\eta\varepsilon}(x) = \frac{1}{(2\pi)^{d/2} \eta^d} \int e^{\mathcal{L}(x'; y) - \frac{1}{2\eta^2} \|x - x'\|^2} dx', \quad \varepsilon \sim \mathcal{N}(0, I_d), \qquad (24)$$

*where $*$ denotes convolution. If PCS has error at most $\varepsilon_{\mathsf{PCS}}$ in total variation and DDS has error at most $\varepsilon_{\mathsf{DDS}}$ in total variation per iteration, then for any accuracy goal $\varepsilon_{\mathsf{acc}} > 0$, with $K \asymp \frac{\log(1/\varepsilon_{\mathsf{acc}})}{1-\lambda}$, we have*

$$\mathsf{TV}(p_{\widehat{x}_K}, \pi_\eta) \lesssim \varepsilon_{\mathsf{acc}} \sqrt{\chi^2(p_{\widehat{x}_1} \| \pi_\eta)} + \frac{1}{1-\lambda} (\varepsilon_{\mathsf{DDS}} + \varepsilon_{\mathsf{PCS}}) \log\left(\frac{1}{\varepsilon_{\mathsf{acc}}}\right). \qquad (25)$$

Before interpreting Theorem 2, we observe that $q_0(x) = e^{\mathcal{L}(x;y)}$, thus $\pi_0(x) \propto p^\star(x) e^{\mathcal{L}(x;y)}$ coincides with the desired posterior distribution $p^\star(\cdot|y)$. Thus Theorem 2 tells us that, assuming a constant annealing schedule $\eta_k = \eta$, the output of DPnP converges in total variation to the distribution $\pi_\eta$, which is a distorted version of the desired posterior distribution up to level $\eta$, with sufficiently many iterations. A few remarks are in order.

**Non-diminishing $\eta$.** It can be seen from Theorem 2 that even with a nonzero $\eta$, DPnP already enforces the data prior strictly. On the other hand, the measurement consistency is distorted by an order of $\eta$. This is usually tolerable, since the measurements are themselves contaminated by

Table 1: Samples of different algorithms for phase retrieval, where DPnP generate images of higher quality and recover fine details of the image more faithfully than the state-of-the-art DPS [CKM+23] and LGD-MC [SZY+23] algorithms. Pixel-based ReSample [SKZ+23] did not generate meaningful images, possibly due to high nonlinearity of phase retrieval.

Table 2: Samples of different algorithms for quantized sensing.

noise, thus when $\eta$ is smaller than the noise level, the distortion would be tolerable. In practice, it is beneficial to choose an annealing schedule of $\{\eta_k\}$, which will be elaborated in Section 5.

**Provable robustness.** Theorem 2 indicates the performance of DPnP degenerates gracefully in the presence of sampling errors. To the best of our knowledge, this is the first provably consistent and robust posterior sampling method for nonlinear inverse problems using score-based diffusion priors.

## 5 Numerical experiments

We provide preliminary numerical evidence to corroborate the promise of DPnP in solving both linear and nonlinear image reconstruction tasks. We denote DPnP with the subroutines DDS-DDPM and DDS-DDIM as DPnP-DDPM and DPnP-DDIM respectively.

Table 3: Evaluation of solving inverse problems on FFHQ $256 \times 256$ validation dataset (1k samples). Despite considerable efforts to optimize parameters, pixel-based ReSample did not generate meaningful results for phase retrieval.

| Algorithm | Super-resolution (4x, linear) | | Phase retrieval (nonlinear) | | Quantized sensing (nonlinear) | | Time per sample |
|---|---|---|---|---|---|---|---|
| | LPIPS ↓ | PSNR ↑ | LPIPS ↓ | PSNR ↑ | LPIPS ↓ | PSNR ↑ | |
| DPnP-DDIM (ours) | **0.301** | **24.2** | **0.376** | **22.4** | **0.293** | **24.2** | ∼ 90s |
| DPS [CKM+23] | 0.331 | 23.1 | 0.490 | 17.4 | 0.367 | 21.7 | ∼ 60s |
| LGD-MC ($n = 5$) [SZY+23] | 0.318 | 23.9 | 0.522 | 16.4 | 0.317 | 23.9 | ∼ 60s |
| ReSample (pixel-based) [SKZ+23] | 0.313 | 23.9 | - | - | 0.318 | 22.6 | ∼ 70s |

Table 4: Evaluation of solving inverse problems on ImageNet $256 \times 256$ validation dataset (1k samples). Despite considerable efforts to optimize parameters, pixel-based ReSample did not generate meaningful results for phase retrieval.

| Algorithm | Super-resolution (4x, linear) | | Phase retrieval (nonlinear) | | Quantized sensing (nonlinear) | | Time per sample |
|---|---|---|---|---|---|---|---|
| | LPIPS ↓ | PSNR ↑ | LPIPS ↓ | PSNR ↑ | LPIPS ↓ | PSNR ↑ | |
| DPnP-DDIM (ours) | **0.416** | **21.6** | **0.562** | **13.4** | **0.363** | **23.0** | $\sim 240$s |
| DPS [CKM$^+$23] | 0.473 | 20.2 | 0.677 | **13.4** | 0.542 | 18.7 | $\sim 150$s |
| LGD-MC ($n = 5$) [SZY$^+$23] | **0.416** | 20.9 | 0.592 | 12.8 | 0.384 | 22.3 | $\sim 150$s |
| ReSample (pixel-based) [SKZ$^+$23] | 0.464 | 20.1 | - | - | 0.414 | 19.8 | $\sim 180$s |

**Experimental setups.** We compare DPnP with the state-of-the-art algorithms including DPS [CKM$^+$23], LGD-MC [SZY$^+$23], and pixel-based ReSample [SKZ$^+$23] for super-resolution (linear), phase retrieval (nonlinear), and quantized sensing (nonlinear). Definitions of these forward measurement models are in Appendix G.2. The annealing schedule $\{\eta_k\}$ of DPnP is fixed across *all* tasks (Appendix H.2), while DPS, LGD-MC, and ReSample are fine-tuned with reasonable effort for best performance. All experiments are run on a single Nvidia L40 GPU. More details and experiments are in Appendix G.

**Sample images.** We present the sample images of different algorithms for the most complicated task of phase retrieval. For phase retrieval, Fourier transform is performed to the image with a coded mask, and only the magnitude of the Fourier transform is taken as the measurement [SEC$^+$15]. Due to the nonlinearity of the operation of taking magnitude, the forward model is nonlinear. The samples generated by different algorithms are shown in Table 1.

We also present the sample images for the nonlinear problem of quantized sensing. In quantized sensing, each pixel of the image is randomly dithered and then quantized to one bit per channel. Nonlinearity of quantizing renders this forward model nonlinear. The samples generated by different algorithms are shown in Table 2.

**Evaluation.** We evaluate the performance of DPnP on the FFHQ validation dataset [KLA19] and the ImageNet validation dataset [RDS$^+$15]. Since DPnP-DDIM has similar performance with DPnP-DDPM but admits much faster implementation, only DPnP-DDIM is evaluated. The LPIPS and PSNR are shown in Table 3 and Table 4. These two metrics are arguably the more relevant ones for solving inverse problems. For comparison under other metrics such as FID, SSIM, cf. Appendix G.4.

It can be seen that, DPnP is capable of solving both linear and nonlinear problems, and, in comparison with prior state-of-the-art, performs better in recovering fine and crisper details.

## 6 Discussion

This paper sets forth a rigorous and versatile algorithmic framework called DPnP for solving nonlinear inverse problems via posterior sampling, using image priors prescribed by score-based diffusion models with general forward models. DPnP alternates between two sampling steps implemented by DDS and PCS, to promote consistency with the data prior constraint and the measurement constraint respectively. We provide both asymptotic and non-asymptotic convergence guarantees, establishing DPnP as the first provably consistent and robust score-based diffusion posterior sampling method for general nonlinear inverse problems.

## Acknowledgments and Disclosure of Funding

This work is supported in part by Office of Naval Research under N00014-19-1-2404, and by National Science Foundation under DMS-2134080 and ECCS-2126634. X. Xu is also gratefully supported by the Axel Berny Presidential Graduate Fellowship at Carnegie Mellon University.

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

## A   Related works

**Algorithmic unrolling and plug-and-play image reconstruction.** Composite optimization algorithms, which aim to minimize the sum of a measurement fidelity term and a regularization term promoting desirable solution structures, have been the backbone of inverse problem solvers. To unleash the power of deep learning, [GL10] advocates the perspective of algorithmic unrolling, which turns an iterative algorithm into concatenations of linear and nonlinear layers like in a neural network. [VBW13] recognized that the proximal mapping step in many composite optimization algorithms can be regarded as a denoiser or denoising operator with respect to the given prior, and proposed to "plug in" alternative denoisers, in particular state-of-the-art deep learning denoisers, leading to a class of popular algorithms known as plug-and-play methods [BCSB18]; see [MLE21] for a review.

**Regularization by denoising and score matching.** [Vin11] pointed out a connection between score matching and image denoising, which is a consequence of the Tweedie's formula [Efr11]. The regularization by denoising (RED) framework [REM17] follows the plug-and-play framework to minimize a regularized objective function, where the regularizer is defined based on the plug-in image denoiser; [RS18] later clarified that the RED framework can be interpreted as score matching by denoising using the Tweedie's formula. [KVE21] developed a stochastic image denoiser for posterior sampling of image denoising using annealed Langevin dynamics. [FBS24] provided a framework to learn exact proximal operators for inverse problems.

**Plug-and-play posterior sampling.** Motivated by the need to characterize the uncertainty, tackling image reconstruction as posterior sampling from a Bayesian perspective is another important approach. Our method is inspired by the plug-and-play framework but takes on a sampling perspective, exploiting the connection between optimization and sampling [Wib18]. Along similar lines, [LBA+22, BB23] proposed Bayesian counterparts of plug-and-play for posterior sampling, where they leveraged the connection to score matching for sampling from the image prior, but did not consider score-based diffusion models for the image prior, which is a key aspect of ours; see also [SWC+23]. [CDC23] extended the split Gibbs sampler [VDC19] in the plug-and-play framework, and advocated the use of score-based diffusion models such as DDPM [HJA20] for image denoising based on heuristic observations. In contrast, we rigorously derive the denoising diffusion samplers from first principles, unraveling critical gaps from naïve applications of the generative samplers to denoising, and offer theoretical guarantees on the correctness of our approach.

**Score-based diffusion models as image priors.** Several representative methods for solving inverse problems using score-based diffusion priors alternates between taking steps along the diffusion process and projecting onto the measurement constraint, e.g., [CKM+23, KEES22, SKZ+23, CLY23, GMJS22, SVMK22]. However, these approaches do not possess asymptotic consistency guarantees. [SZY+23] proposed to use multiple Monte Carlo samples to reduce bias. On the other hand, [CICM23] developed Monte Carlo guided diffusion methods for Bayesian linear inverse problems which tend to be computationally expensive, and [DS24] recently introduced a filtering perspective and applied particle filtering. Although asymptotically consistent, these approaches are limited to linear inverse problems. [TYT+22, WTN+23] introduced sequential Monte Carlo (SMC) algorithms for conditional sampling using unconditional diffusion models that are asymptotically exact. [MSKV24] developed a variational perspective that connects to the regularization by denoising framework. [GJP+24] showed that the worst-case complexity of diffusion posterior sampling can take super-polynomial time regardless of the algorithm in use.

**Theory of diffusion models and score matching.** A number of recent papers have studied the non-asymptotic convergence rates of popular diffusion samplers, including but not limited to stochastic DDPM-type samplers [CCL+22, CLL23, LWCC23, BDBDD24, TZ24], deterministic DDIM-type samplers [LWCC23, CCL+23, CDD23], and accelerated samplers [LHE+24, LHW24]. In addition, the statistical efficiency of score matching has also been investigated [KHR23, PRS+24].

## B   Discrete-time formulation of diffusion processes

The discrete-time forward and backward diffusion processes can be understood as two processes constructed in the following manner:

1) a forward process

$$x_0 \to x_1 \to \cdots \to x_T$$

that starts with samples from the target image distribution and diffuses into a noise distribution (e.g., standard Gaussians) by gradually injecting noise into the samples;

2) a reverse process
$$x_T^{\mathsf{rev}} \to x_{T-1}^{\mathsf{rev}} \to \cdots \to x_0^{\mathsf{rev}}$$
that starts from pure noise (e.g., standard Gaussians) and converts it into samples whose distribution is close to the target image distribution.

Consider the forward Markov process in $\mathbb{R}^d$ that starts with a sample from the data distribution $p_X$, and adds noise over the trajectory according to

$$x_0 \sim p^\star, \tag{26a}$$
$$x_t = \sqrt{1 - \beta_t}\, x_{t-1} + \sqrt{\beta_t}\, w_t, \qquad 1 \le t \le T, \tag{26b}$$

where $\{w_t\}_{1 \le t \le T}$'s are independent standard Gaussian vectors, i.e., $w_t \overset{\text{i.i.d.}}{\sim} \mathcal{N}(0, I_d)$, and $\{\beta_t \in (0, 1)\}$ describes the noise-injection rates used in each step. Therefore, we can write $x_t$ equivalently as

$$x_t := \sqrt{\bar{\alpha}_t}\, x_0 + \sqrt{1 - \bar{\alpha}_t}\, \varepsilon_t, \quad \varepsilon_t \sim \mathcal{N}(0, I_d), \quad t = 0, 1, \cdots, T. \tag{27}$$

Here, $(\bar{\alpha}_t)_{t=0,1,\cdots,T}$ is the *schedule* of diffusion given by

$$\alpha_t := 1 - \beta_t, \qquad \bar{\alpha}_t := \prod_{k=1}^{t} \alpha_k, \qquad 1 \le t \le T. \tag{28}$$

Clearly, it verifies that $1 \ge \bar{\alpha}_0 > \bar{\alpha}_1 > \cdots > \bar{\alpha}_T > 0$. As long as $\bar{\alpha}_T$ is vanishing, it is easy to observe that the distribution of $x_T$ approaches $\mathcal{N}(0, I_d)$.

**Score functions.** As will be seen, in order to sample from $p^\star$, it turns out to be sufficient to learn the score functions of $p_{x_t}$ at each step of the forward process, defined as

$$s_t^\star(x) = \nabla \log p_{x_t}(x), \qquad t = 0, 1, \cdots, T. \tag{29}$$

As in the continuous time, the score function can be viewed as a MMSE estimate:

$$s_t^\star(x) = -\frac{1}{\sqrt{1 - \bar{\alpha}_t}} \underbrace{\mathbb{E}_{x_0 \sim p^\star,\, \varepsilon_t \sim \mathcal{N}(0, I_d)}\big(\varepsilon_t \mid \sqrt{\bar{\alpha}_t} x_0 + \sqrt{1 - \bar{\alpha}_t}\varepsilon_t = x\big)}_{=: \varepsilon_t^\star(x)}. \tag{30}$$

Comparing (27) and (2), it can be checked that the discrete-time diffusion process can be embedded into the continuous-time one via the time change

$$t \mapsto \frac{1}{2} \log \frac{1}{\bar{\alpha}_t},$$

in the sense that

$$x_t^\star \overset{\text{(d)}}{=} X_{\frac{1}{2} \log \frac{1}{\bar{\alpha}_t}}.$$

This establishes a correspondence between the continuous-time formulation and the discrete-time formulation.

Within the discrete-time formulation, we assume the score function estimates are given as $\widehat{s}_t(\cdot) : \mathbb{R}^d \to \mathbb{R}^d$, $t = 1, \ldots, T$ such that $\widehat{s}_t \approx s_t^\star$ in analogy with the continuous-time counterpart.

## C  Details of algorithm subroutines

With the discrete-time perspective established in Appendix B, we are now ready to present the detailed description and the implementation of our algorithms DDS-DDPM (Algorithm 2), DDS-DDIM (Algorithm 3), and PCS (Algorithm 4).

## D  Score functions of diffusion denoising samplers

### D.1  Proof of Lemma 1

*Proof.* The marginal distribution (14) of the heat flow can be written as

$$Y_\tau \overset{\text{(d)}}{=} Y_0 + \sqrt{\tau}\varepsilon, \quad Y_0 \sim p^\star, \; \varepsilon \sim \mathcal{N}(0, I_d). \tag{31}$$

Comparing (2) and (31), it is not hard to check that

$$Y_\tau \overset{\text{(d)}}{=} \sqrt{1 + \tau} X_{\frac{1}{2} \log(1+\tau)}.$$

---

**Algorithm 2** Denoising Diffusion Sampler (stochastic) DDS-DDPM$(x_{\mathsf{noisy}}, \widehat{s}, \eta)$

---

**Input**: noisy data $x_{\mathsf{noisy}} \in \mathbb{R}^d$, score estimates $\widehat{s} := \{\widehat{s}_t(\cdot) : \mathbb{R}^d \to \mathbb{R}^d, t = 1, \dots, T\}$ or noise estimates $\widehat{\varepsilon} = \{\widehat{\varepsilon}_t(\cdot) : \mathbb{R}^d \to \mathbb{R}^d, t = 1, \dots, T\}$, and noise level $\eta > 0$.
**Scheduling**: Compute the diffusion schedule $(\tau_t)_{0 \le t \le T'}$ by

$$\tau_t = \bar{\alpha}_t^{-1} - 1, \quad 0 \le t \le T',$$

where

$$T' := \max\left\{t : 0 \le t \le T, \bar{\alpha}_t > \frac{1}{\eta^2 + 1}\right\}.$$

**Initialization**: Set $\widehat{x}_{T'} = x_{\mathsf{noisy}}$.
**Diffusion**: **for** $t = T', T' - 1, \dots, 1$ **do**

$$\widehat{x}_{t-1} = \widehat{x}_t - 2(\sqrt{\tau_t} - \sqrt{\tau_{t-1}})\,\widehat{\varepsilon}_t + \sqrt{\tau_t - \tau_{t-1}}\,w_t, \quad w_t \sim \mathcal{N}(0, I_d).$$

where

$$\widehat{\varepsilon}_t := \widehat{\varepsilon}_t(\sqrt{\bar{\alpha}_t}\,\widehat{x}_t) = -\frac{1}{\sqrt{1 - \bar{\alpha}_t}}\widehat{s}_t\left(\sqrt{\bar{\alpha}_t}\,\widehat{x}_t\right).$$

**Output**: $\widehat{x}_0$.

---

**Algorithm 3** Denoising Diffusion Sampler (deterministic) DDS-DDIM$(x_{\mathsf{noisy}}, \widehat{s}, \eta)$

---

**Input**: noisy data $x_{\mathsf{noisy}} \in \mathbb{R}^d$, score estimates $\widehat{s} := \{\widehat{s}_t(\cdot) : \mathbb{R}^d \to \mathbb{R}^d, t = 1, \dots, T\}$ or noise estimates $\widehat{\varepsilon} = \{\widehat{\varepsilon}_t(\cdot) : \mathbb{R}^d \to \mathbb{R}^d, t = 1, \dots, T\}$, and noise level $\eta > 0$.
**Scheduling**: Compute the diffusion schedule $(\bar{u}_t)_{0 \le t \le T'}$ by

$$\bar{u}_t = \frac{(\eta^2 + 1)\bar{\alpha}_t - 1}{\eta^2 + \bar{\alpha}_t - 1}, \quad 0 \le t \le T',$$

where

$$T' := \max\left\{t : 0 \le t \le T, \bar{\alpha}_t > \frac{1}{\eta^2 + 1}\right\}.$$

**Initialization**: Draw $z_{T'} \sim \mathcal{N}(0, I_d)$.
**Diffusion**: **for** $t = T', T' - 1, \dots, 1$ **do**

$$z_{t-1} = \frac{\sqrt{(\eta^2 - 1)\bar{u}_{t-1} + 1}}{\sqrt{(\eta^2 - 1)\bar{u}_t + 1}} z_t + \sqrt{(\eta^2 - 1)\bar{u}_{t-1} + 1} \cdot \big(h(\eta, \bar{u}_{t-1}) - h(\eta, \bar{u}_t)\big)\widehat{\varepsilon}_t,$$

where

$$h(\eta, u) := -\arctan\frac{\eta}{\sqrt{u^{-1} - 1}},$$
$$\widehat{\varepsilon}_t := \widehat{\varepsilon}_t\left(\sqrt{\bar{\alpha}_t}x_{\mathsf{noisy}} + \frac{\eta^2\sqrt{\bar{u}_t\bar{\alpha}_t}z_t}{(\eta^2 - 1)\bar{u}_t + 1}\right) = -\frac{1}{\sqrt{1 - \bar{\alpha}_t}}\widehat{s}_t\left(\sqrt{\bar{\alpha}_t}x_{\mathsf{noisy}} + \frac{\eta^2\sqrt{\bar{u}_t\bar{\alpha}_t}z_t}{(\eta^2 - 1)\bar{u}_t + 1}\right).$$

**Output**: $x_{\mathsf{noisy}} + z_0$.

---

Denote $\theta = \frac{1}{2}\log(1 + \tau)$ as a short-hand. We have

$$p_{Y_\tau}(x) = p_{\sqrt{1+\tau}X_\theta}(x) \propto p_{X_\theta}\left(\frac{1}{\sqrt{1+\tau}}x\right).$$

Therefore it follows that

$$\nabla \log p_{Y_\tau}(x) = \nabla_x \log p_{X_\theta}\left(\frac{1}{\sqrt{1+\tau}}x\right) = \frac{1}{\sqrt{1+\tau}}s\left(\theta, \frac{1}{\sqrt{1+\tau}}x\right),$$

where we used the definition $s(\theta, \cdot) = \nabla \log p_{X_\theta}(\cdot)$. Plugging the definition $\theta = \frac{1}{2}\log(1 + \tau)$ into the above equation yields the desired result. $\qquad\square$

**Algorithm 4** Proximal Consistency Sampler $\mathsf{PCS}(x, y, \mathcal{L}, \eta)$ (adapted from Metropolis-Adjusted Langevin Algorithm [RR98])

---

**Input**: starting point $x \in \mathbb{R}^d$, measurements $y \in \mathbb{R}^m$, log-likelihood function of the forward model $\mathcal{L}(\cdot; y)$, proximal parameter $\eta > 0$.
**Hyperparameter**: Langevin stepsize $\gamma$, and the number of iterations $N$.
**Initialization**: $z_0 = x$.
**Update**: **for** $n = 0, 1, \cdots, N-1$ **do**

    (1) **One step of discretized Langevin**: Set $r = \mathrm{e}^{-\gamma/\eta^2}$, and

$$z_{n+\frac{1}{2}} = rz_n + (1-r)x + \eta^2(1-r)\nabla_{z_n}\mathcal{L}(z_n; y) + \eta\sqrt{1-r^2}w_n, \quad w_n \sim \mathcal{N}(0, I_d).$$

    This is equivalent to drawing $z_{n+\frac{1}{2}}$ from a distribution with density $Q(\cdot; z_n)$, where

$$Q(z'; z) = \frac{1}{(2\pi(1-r^2))^{d/2}} \exp\left(-\frac{\left\|z' - \left(rz + (1-r)x + \eta^2(1-r)\nabla_z\mathcal{L}(z; y)\right)\right\|^2}{2(1-r^2)}\right).$$

    (2) **Metropolis adjustment**: Compute

$$q = \frac{\exp\left(\mathcal{L}(z_{n+\frac{1}{2}}; y) - \frac{1}{2\eta^2}\|z_{n+\frac{1}{2}} - x\|^2\right)}{\exp\left(\mathcal{L}(z_n; y) - \frac{1}{2\eta^2}\|z_n - x\|^2\right)} \cdot \frac{Q(z_n; z_{n+\frac{1}{2}})}{Q(z_{n+\frac{1}{2}}; z_n)},$$

    and set

$$z_{n+1} = \begin{cases} z_{n+\frac{1}{2}}, & \text{with probability } \min(1, q), \\ z_n & \text{with probability } 1 - \min(1, q). \end{cases}$$

**Output**: $z_N$.

---

### D.2 Proof of Lemma 2

*Proof.* We first compute the probability density function of $z$. Recall that $z = w - x_{\mathsf{noisy}}$, thus applying Bayes rule yields

$$p_z(x) = p_w(x + x_{\mathsf{noisy}}) = p^\star(x^\star = x + x_{\mathsf{noisy}}|x^\star + \xi = x_{\mathsf{noisy}})$$
$$= \frac{p^\star(x + x_{\mathsf{noisy}})p_\xi(-x)}{p_{x^\star + \xi}(x_{\mathsf{noisy}})} \propto p^\star(x + x_{\mathsf{noisy}})p_\xi(-x),$$

where $\xi \sim \mathcal{N}(0, \eta^2 I_d)$. It is straightforward to compute

$$p_\xi(-x) = \frac{1}{(2\pi)^{d/2}\eta^d}\mathrm{e}^{-\frac{1}{2\eta^2}\|x\|^2},$$

therefore

$$p_z(x) \propto p^\star(x + x_{\mathsf{noisy}})\mathrm{e}^{-\frac{1}{2\eta^2}\|x\|^2}. \tag{32}$$

We proceed to compute the probability density function of $Z_\tau$. According to (19), it follows that
$$p_{Z_\tau}(x) = p_{\mathrm{e}^{-\tau}z} * p_{\sqrt{1-\mathrm{e}^{-2\tau}}\varepsilon}(x)$$

$$= \int p_{\mathrm{e}^{-\tau}z}(x') \, p_{\sqrt{1-\mathrm{e}^{-2\tau}}\varepsilon}(x - x')\mathrm{d}x'$$

$$\propto \int p_z(\mathrm{e}^\tau x') \exp\left(-\frac{1}{2(1-\mathrm{e}^{-2\tau})}\|x - x'\|^2\right)\mathrm{d}x'$$

$$\propto \int p^\star(x_{\mathsf{noisy}} + \mathrm{e}^\tau x') \exp\left(-\frac{1}{2\eta^2}\|\mathrm{e}^\tau x'\|^2\right) \exp\left(-\frac{1}{2(1-\mathrm{e}^{-2\tau})}\|x - x'\|^2\right)\mathrm{d}x',$$

$$\propto \int p^\star(x') \exp\left(-\frac{1}{2\eta^2}\|x' - x_{\mathsf{noisy}}\|^2\right) \exp\left(-\frac{1}{2(1-\mathrm{e}^{-2\tau})}\|x - \mathrm{e}^{-\tau}(x' - x_{\mathsf{noisy}})\|^2\right)\mathrm{d}x',$$
$$\tag{33}$$

where $*$ denotes convolution, the penultimate line follows from (32) and the last line follow from the change of variable $x' \mapsto \mathrm{e}^{-\tau}(x' - x_{\mathsf{noisy}})$. One may exercise some brute force to verify that

$$\exp\left(-\frac{1}{2\eta^2}\|x' - x_{\mathsf{noisy}}\|^2\right) \exp\left(-\frac{1}{2(1-\mathrm{e}^{-2\tau})}\|x - \mathrm{e}^{-\tau}(x' - x_{\mathsf{noisy}})\|^2\right)$$

$$= \exp\left(-\frac{e^{2\tau}\|x\|^2}{2(\eta^2 + e^{2\tau} - 1)}\right)\exp\left(-\frac{1}{2(1 - e^{-2\tilde{\tau}})}\left\|e^{-\tilde{\tau}}x_{\mathsf{noisy}} + \frac{e^{\tau - \tilde{\tau}}\eta^2 x}{\eta^2 + e^{2\tau} - 1} - e^{-\tilde{\tau}}x'\right\|^2\right)$$

$$\propto \exp\left(-\frac{e^{2\tau}\|x\|^2}{2(\eta^2 + e^{2\tau} - 1)}\right) p_{\sqrt{1 - e^{-2\tilde{\tau}}}\varepsilon}\left(e^{-\tilde{\tau}}x_{\mathsf{noisy}} + \frac{e^{\tau - \tilde{\tau}}\eta^2 x}{\eta^2 + e^{2\tau} - 1} - e^{-\tilde{\tau}}x'\right),$$

where $\tilde{\tau}$ is as defined in (22). Plug this back into (33), we see

$$p_{Z_\tau}(x) \propto \exp\left(-\frac{e^{2\tau}\|x\|^2}{2(\eta^2 + e^{2\tau} - 1)}\right)\int p^\star(x') p_{\sqrt{1 - e^{-2\tilde{\tau}}}\varepsilon}\left(e^{-\tilde{\tau}}x_{\mathsf{noisy}} + \frac{e^{\tau - \tilde{\tau}}\eta^2 x}{\eta^2 + e^{2\tau} - 1} - e^{-\tilde{\tau}}x'\right)\mathrm{d}x'$$

$$\propto \exp\left(-\frac{e^{2\tau}\|x\|^2}{2(\eta^2 + e^{2\tau} - 1)}\right)\int p^\star(e^{\tilde{\tau}}x') p_{\sqrt{1 - e^{-2\tilde{\tau}}}\varepsilon}\left(e^{-\tilde{\tau}}x_{\mathsf{noisy}} + \frac{e^{\tau - \tilde{\tau}}\eta^2 x}{\eta^2 + e^{2\tau} - 1} - x'\right)\mathrm{d}x'$$

$$\propto \exp\left(-\frac{e^{2\tau}\|x\|^2}{2(\eta^2 + e^{2\tau} - 1)}\right) p_{e^{-\tau}x_0} * p_{\sqrt{1 - e^{-2\tilde{\tau}}}\varepsilon}\left(e^{-\tilde{\tau}}x_{\mathsf{noisy}} + \frac{e^{\tau - \tilde{\tau}}\eta^2 x}{\eta^2 + e^{2\tau} - 1}\right)$$

$$\propto \exp\left(-\frac{e^{2\tau}\|x\|^2}{2(\eta^2 + e^{2\tau} - 1)}\right) p_{X_{\tilde{\tau}}}\left(e^{-\tilde{\tau}}x_{\mathsf{noisy}} + \frac{e^{\tau - \tilde{\tau}}\eta^2 x}{\eta^2 + e^{2\tau} - 1}\right),$$

where the second line applies the change of variable $x' \mapsto e^{\tilde{\tau}}x'$ in the integral, the penultimate line follows from $p_{e^{-\tilde{\tau}}x_0}(x') \propto p^\star(e^{\tilde{\tau}}x')$ (since $x_0 \sim p^\star$), and the last line follows from $X_{\tilde{\tau}} \overset{(\mathrm{d})}{=} e^{-\tilde{\tau}}x_0 + \sqrt{1 - e^{-2\tilde{\tau}}}\varepsilon$.

Finally, from the above formula, we obtain

$$\nabla \log p_{Z_\tau}(x) = \nabla_x\left(-\frac{e^{2\tau}\|x\|^2}{2(\eta^2 + e^{2\tau} - 1)}\right) + \nabla_x \log p_{X_{\tilde{\tau}}}\left(e^{-\tilde{\tau}}x_{\mathsf{noisy}} + \frac{e^{\tau - \tilde{\tau}}\eta^2 x}{\eta^2 + e^{2\tau} - 1}\right)$$

$$= -\frac{e^{2\tau}x}{\eta^2 + e^{2\tau} - 1} + \frac{e^{\tau - \tilde{\tau}}\eta^2}{\eta^2 + e^{2\tau} - 1}s\left(\tilde{\tau}, e^{-\tilde{\tau}}x_{\mathsf{noisy}} + \frac{e^{\tau - \tilde{\tau}}\eta^2 x}{\eta^2 + e^{2\tau} - 1}\right),$$

where we used the definition $s(\tilde{\tau}, \cdot) = \nabla \log p_{X_{\tilde{\tau}}}(\cdot)$. $\qquad\square$

# E Discretization via the exponential integrator

## E.1 General form of the exponential integrator

Consider a SDE of the form:

$$\mathrm{d}M_\tau = \big(v(\tau)M_\tau + f(\tau, M_\tau)\big)\mathrm{d}\tau + \sqrt{\beta}\mathrm{d}B_\tau, \quad \tau \in [0, \tau_\infty], \quad M_0 \sim p_{M_0},$$

where $v : [0, \tau_\infty] \to \mathbb{R}$, $f : [0, \tau_\infty] \times \mathbb{R}^d \to \mathbb{R}^d$ are deterministic functions, and $\beta > 0$ is a constant. Given discretization time points $0 = \tau_0 \le \tau_1 \le \cdots \le \tau_k \le \tau_\infty$, a naïve way to discretize the SDE is

$$M_{\tau_{i+1}} - M_{\tau_i} \approx \big(v(\tau_i)M_{\tau_i} + f(\tau_i, M_{\tau_i})\big)(\tau_{i+1} - \tau_i) + \sqrt{\beta}\sqrt{\tau_{i+1} - \tau_i}\varepsilon_i, \quad i = 0, 1, \cdots, k - 1,$$

where $\varepsilon_i \sim \mathcal{N}(0, I_d)$ is a standard $d$-dimensional Gaussian random vector which is independent of $M_{\tau_i}$. Although this approach is straightforward, it has the drawback that the linear term $v(\tau)M_\tau$ is discretized rather crude. For example, for the OU process where $v \equiv -1$, $f \equiv 0$, $\beta = 2$, the SDE can be solved analytically as in (2), while the above approach still has a discretization error.

A more accurate discretization, known to significantly improve the quality of score-based generative models, is given by the *exponential integrator* [ZC22], which preserves the linear term and discretizes the SDE to

$$\mathrm{d}\widehat{M}_\tau = \big(v(\tau)\widehat{M}_\tau + f(\tau_i, \widehat{M}_{\tau_i})\big)\mathrm{d}\tau + \sqrt{\beta}\mathrm{d}B_\tau, \quad \tau \in [\tau_i, \tau_{i+1}], \quad i = 0, 1, \cdots, k,$$

with initialization $\widehat{M}_0 \sim p_{M_0}$. On each time interval $[\tau_i, \tau_{i+1}]$, this is simply a linear SDE, which can be explicitly solved by

$$\widehat{M}_\tau \overset{(\mathrm{d})}{=} e^{V(\tau) - V(\tau_i)}\widehat{M}_{\tau_i} + \left(\int_{\tau_i}^\tau e^{V(\tau) - V(\tilde{\tau})}\mathrm{d}\tilde{\tau}\right)f(\tau_i, \widehat{M}_{\tau_i}) + \sqrt{\beta}\left(\int_{\tau_i}^\tau e^{2(V(\tau) - V(\tilde{\tau}))}\mathrm{d}\tilde{\tau}\right)^{1/2}\varepsilon_i,$$

where $V$ is the antiderivative of $v$:

$$V(\tau) = \int_0^\tau v(\tilde{\tau})\mathrm{d}\tilde{\tau}.$$

Taking $\tau = \tau_{i+1}$, we obtain

$$\widehat{M}_{\tau_{i+1}} \stackrel{(d)}{=} e^{V(\tau_{i+1})-V(\tau_i)}\widehat{M}_{\tau_i} + \left(\int_{\tau_i}^{\tau_{i+1}} e^{V(\tau_{i+1})-V(\tilde{\tau})}d\tilde{\tau}\right) f(\tau_i, \widehat{M}_{\tau_i})$$
$$+ \sqrt{\beta}e^{V(\tau_{i+1})}\left(\int_{\tau_i}^{\tau_{i+1}} e^{2(V(\tau_{i+1})-V(\tilde{\tau}))}d\tilde{\tau}\right)^{1/2}\varepsilon_i, \tag{34}$$

which provides an iterative formula to compute $\widehat{M}_{\tau_{i+1}}$.

### E.2 Discretization of DDS-DDPM

Plug the expression of $\nabla \log p_{Y_\tau}$ in Lemma 1 into (16), and use the notation $\tau^{\text{rev}} = \eta^2 - \tau$, we obtain, for $\tau \in [0, \eta^2]$, that

$$dY_{\tau^{\text{rev}}}^{\text{rev}} = \frac{1}{\sqrt{1+\tau^{\text{rev}}}}s\left(\frac{1}{2}\log(1+\tau^{\text{rev}}), \frac{Y_{\tau^{\text{rev}}}^{\text{rev}}}{\sqrt{1+\tau^{\text{rev}}}}\right)d\tau + d\tilde{B}_\tau$$
$$= -\frac{1}{\sqrt{\tau^{\text{rev}}}}\varepsilon^{\text{cont}}\left(\frac{1}{2}\log(1+\tau^{\text{rev}}), \frac{Y_{\tau^{\text{rev}}}^{\text{rev}}}{\sqrt{1+\tau^{\text{rev}}}}\right)d\tau + d\tilde{B}_\tau.$$

**Choosing discretization time points.** To discretize this SDE, we first choose the discretization time points. Recalling (**??**), it is most reasonable to discretize at those time points $0 \leq \tau_0^{\text{rev}} \leq \cdots \leq \tau_{T'}^{\text{rev}} \leq \eta^2$ which satisfy

$$\frac{1}{2}\log(1+\tau_t^{\text{rev}}) = \frac{1}{2}\log\frac{1}{\bar{\alpha}_t}, \quad 0 \leq t \leq T'.$$

This solves to

$$\tau_t^{\text{rev}} = \bar{\alpha}_t^{-1} - 1. \tag{35}$$

The requirement that $\tau_t^{\text{rev}} \leq \eta^2$ translates to $\bar{\alpha}_t \geq \frac{1}{1+\eta^2}$, which yields the following choice of $T'$:

$$T' := \max\left\{t : 0 \leq t \leq T, \bar{\alpha}_t > \frac{1}{\eta^2+1}\right\}. \tag{36}$$

**Applying the exponential integrator.** Now we apply the exponential integrator to discretize the SDE on each time interval $\tau^{\text{rev}} \in [\tau_{t-1}, \tau_t], t = 1, \cdots, T'$ as follows:

$$d\widehat{Y}_{\tau^{\text{rev}}}^{\text{rev}} = -\frac{1}{\sqrt{\tau^{\text{rev}}}}\varepsilon^{\text{cont}}\left(\frac{1}{2}\log(1+\tau_t^{\text{rev}}), \frac{\widehat{Y}_{\tau_t^{\text{rev}}}^{\text{rev}}}{\sqrt{1+\tau_t^{\text{rev}}}}\right)d\tau + d\tilde{B}_\tau,$$
$$= -\frac{1}{\sqrt{\tau^{\text{rev}}}}\varepsilon_t^\star\left(\frac{\widehat{Y}_{\tau_t^{\text{rev}}}^{\text{rev}}}{\sqrt{1+\tau_t^{\text{rev}}}}\right)d\tau + d\tilde{B}_\tau$$
$$= -\frac{1}{\sqrt{\tau^{\text{rev}}}}\varepsilon_t^\star\left(\sqrt{\bar{\alpha}_t}\widehat{Y}_{\tau_t^{\text{rev}}}^{\text{rev}}\right)d\tau + d\tilde{B}_\tau.$$

The SDE can be integrated directly on $\tau^{\text{rev}} \in [\tau_{t-1}, \tau_t]$ (see also (34), with $v \equiv 0$), yielding

$$\widehat{Y}_{\tau_{t-1}^{\text{rev}}}^{\text{rev}} = \widehat{Y}_{\tau_t^{\text{rev}}}^{\text{rev}} - 2(\sqrt{\tau_t^{\text{rev}}} - \sqrt{\tau_{t-1}^{\text{rev}}}) \cdot \varepsilon_t^\star\left(\sqrt{\bar{\alpha}_t}\widehat{Y}_{\tau_t^{\text{rev}}}^{\text{rev}}\right) + \int_{\eta^2-\tau_t}^{\eta^2-\tau_{t-1}} d\tilde{B}_\tau d\tau$$
$$\stackrel{(d)}{=} \widehat{Y}_{\tau_t^{\text{rev}}}^{\text{rev}} - 2(\sqrt{\tau_t^{\text{rev}}} - \sqrt{\tau_{t-1}^{\text{rev}}}) \cdot \varepsilon_t^\star\left(\sqrt{\bar{\alpha}_t}\widehat{Y}_{\tau_t^{\text{rev}}}^{\text{rev}}\right) + \sqrt{\tau_t^{\text{rev}} - \tau_{t-1}^{\text{rev}}}w_t,$$

where $w_t \sim \mathcal{N}(0, I_d)$ is independent of $\widehat{Y}_{\tau_t^{\text{rev}}}^{\text{rev}}$. Set $\widehat{x}_t = \widehat{Y}_{\tau_t^{\text{rev}}}^{\text{rev}}$, we obtain

$$\widehat{x}_{t-1} \stackrel{(d)}{=} \widehat{x}_t - 2(\sqrt{\tau_t} - \sqrt{\tau_{t-1}}) \cdot \varepsilon_t^\star\left(\sqrt{\bar{\alpha}_t}\widehat{x}_t\right) + \sqrt{\tau_t - \tau_{t-1}}w_t, \quad w_t \sim \mathcal{N}(0, I_d), \tag{37}$$

which is exactly the update equation in Algorithm 2, except that $\varepsilon_t^\star$ is replaced by the noise estimate $\widehat{\varepsilon}_t$.

### E.3 Discretization of **DDS-DDIM**

Plug in the expression of $s_Z$ in Lemma 2 into the probability flow ODE (20), we obtain

$$
\begin{aligned}
\mathrm{d}Z^{\mathsf{rev}}_{\tau^{\mathsf{rev}}} &= \frac{\eta^2 - 1}{\eta^2 + e^{2\tau^{\mathsf{rev}}} - 1} Z^{\mathsf{rev}}_{\tau^{\mathsf{rev}}} \mathrm{d}\tau + \frac{e^{\tau^{\mathsf{rev}} - \tilde{\tau}(\tau^{\mathsf{rev}})}\eta^2}{\eta^2 + e^{2\tau^{\mathsf{rev}}} - 1} s\left( \tilde{\tau}(\tau^{\mathsf{rev}}), \, e^{-\tilde{\tau}(\tau^{\mathsf{rev}})} x_{\mathsf{noisy}} + \frac{e^{\tau^{\mathsf{rev}} - \tilde{\tau}(\tau^{\mathsf{rev}})}\eta^2 x}{\eta^2 + e^{2\tau^{\mathsf{rev}}} - 1} \right) \mathrm{d}\tau \\
&= \frac{\eta^2 - 1}{\eta^2 + e^{2\tau^{\mathsf{rev}}} - 1} Z^{\mathsf{rev}}_{\tau^{\mathsf{rev}}} \mathrm{d}\tau - \frac{e^{2\tau^{\mathsf{rev}}}}{e^{2\tau^{\mathsf{rev}}} - 1} \varepsilon^{\mathsf{cont}}\left( \tilde{\tau}(\tau^{\mathsf{rev}}), \, e^{-\tilde{\tau}(\tau^{\mathsf{rev}})} x_{\mathsf{noisy}} + \frac{e^{\tau^{\mathsf{rev}} - \tilde{\tau}(\tau^{\mathsf{rev}})}\eta^2 x}{\eta^2 + e^{2\tau^{\mathsf{rev}}} - 1} \right) \mathrm{d}\tau,
\end{aligned}
$$

where the second line used the definition (22).

**Choosing discretization time points.** Similar to the derivation in Appendix E.2, we discretize at time points $0 = \tau^{\mathsf{rev}}_0 \le \tau^{\mathsf{rev}}_1 \le \cdots \le \tau^{\mathsf{rev}}_{T'} \le \eta^2$, which obey

$$
\tilde{\tau}(\tau^{\mathsf{rev}}_t) = \frac{1}{2} \log \frac{1}{\bar{\alpha}_t}, \quad t = 0, 1, \ldots, T', \tag{38}
$$

which solves to

$$
\tau^{\mathsf{rev}}_t = \frac{1}{2} \log \frac{\eta^2 + \bar{\alpha}_t - 1}{(\eta^2 + 1)\bar{\alpha}_t - 1}. \tag{39}
$$

To make this well-defined, we require

$$
\frac{\eta^2 + \bar{\alpha}_t - 1}{(\eta^2 + 1)\bar{\alpha}_t - 1} > 0,
$$

which is equivalent to

$$
\bar{\alpha}_t > \frac{1}{1 + \eta^2}.
$$

This leads to the same choice of $T'$ as in (36). We also set

$$
\tau_\infty = \tau^{\mathsf{rev}}_{T'}.
$$

It is convenient to introduce a notation for the corresponding discrete schedule of $\tau^{\mathsf{rev}}_t$, denoted by

$$
\bar{u}_t = e^{-2\tau^{\mathsf{rev}}_t} = \frac{(\eta^2 + 1)\bar{\alpha}_t - 1}{\eta^2 + \bar{\alpha}_t - 1}, \quad t = 0, 1, \cdots, T'.
$$

**Applying the exponential integrator.** Now we apply the exponential integrator, which discretizes the ODE on each time interval $\tau^{\mathsf{rev}} \in [\tau_{t-1}, \tau_t]$, $t = 1, \cdots, T'$, as

$$
\begin{aligned}
\mathrm{d}\widehat{Z}^{\mathsf{rev}}_{\tau^{\mathsf{rev}}} &= \frac{\eta^2 - 1}{\eta^2 + e^{2\tau^{\mathsf{rev}}} - 1} \widehat{Z}^{\mathsf{rev}}_{\tau^{\mathsf{rev}}} \mathrm{d}\tau - \frac{e^{2\tau^{\mathsf{rev}}}}{e^{2\tau^{\mathsf{rev}}} - 1} \varepsilon^{\mathsf{cont}}\left( \tilde{\tau}(\tau^{\mathsf{rev}}_t), \, e^{-\tilde{\tau}(\tau^{\mathsf{rev}}_t)} x_{\mathsf{noisy}} + \frac{e^{\tau^{\mathsf{rev}}_t - \tilde{\tau}(\tau^{\mathsf{rev}}_t)}\eta^2 \widehat{Z}^{\mathsf{rev}}_{\tau^{\mathsf{rev}}_t}}{\eta^2 + e^{2\tau^{\mathsf{rev}}_t} - 1} \right) \mathrm{d}\tau \\
&= \frac{\eta^2 - 1}{\eta^2 + e^{2\tau^{\mathsf{rev}}} - 1} \widehat{Z}^{\mathsf{rev}}_{\tau^{\mathsf{rev}}} \mathrm{d}\tau - \frac{e^{2\tau^{\mathsf{rev}}}}{e^{2\tau^{\mathsf{rev}}} - 1} \varepsilon^\star_t\left( \sqrt{\bar{\alpha}_t} x_{\mathsf{noisy}} + \frac{e^{\tau^{\mathsf{rev}}_t} \sqrt{\bar{\alpha}_t} \eta^2 \widehat{Z}^{\mathsf{rev}}_{\tau^{\mathsf{rev}}_t}}{\eta^2 + e^{2\tau^{\mathsf{rev}}_t} - 1} \right) \mathrm{d}\tau, \\
&= \frac{\eta^2 - 1}{\eta^2 + e^{2\tau^{\mathsf{rev}}} - 1} \widehat{Z}^{\mathsf{rev}}_{\tau^{\mathsf{rev}}} \mathrm{d}\tau - \frac{e^{2\tau^{\mathsf{rev}}}}{e^{2\tau^{\mathsf{rev}}} - 1} \varepsilon^\star_t\left( \sqrt{\bar{\alpha}_t} x_{\mathsf{noisy}} + \frac{\sqrt{\bar{u}_t} \sqrt{\bar{\alpha}_t} \eta^2 \widehat{Z}^{\mathsf{rev}}_{\tau^{\mathsf{rev}}_t}}{(\eta^2 - 1)\bar{u}_t + 1} \right) \mathrm{d}\tau,
\end{aligned}
$$

where the second line follows from (38), and the last line follows from dividing both the denominator and the numerator in the fraction inside $\widehat{\varepsilon}_t$ by $e^{2\tau^{\mathsf{rev}}_t}$. This is a first-order linear ODE on $\tau^{\mathsf{rev}} \in [\tau_{t-1}, \tau_t]$, which can be solved explicitly (cf. (34)) by

$$
\begin{aligned}
\widehat{Z}^{\mathsf{rev}}_{\tau^{\mathsf{rev}}} &= \frac{\sqrt{(\eta^2 - 1)e^{-2\tau^{\mathsf{rev}}} + 1}}{\sqrt{(\eta^2 - 1)\bar{u}_t + 1}} \widehat{Z}^{\mathsf{rev}}_{\tau^{\mathsf{rev}}_t} \\
&\quad + \sqrt{(\eta^2 - 1)e^{-2\tau^{\mathsf{rev}}} + 1} \cdot \left( h(\eta, e^{-2\tau^{\mathsf{rev}}}) - h(\eta, \bar{u}_t) \right) \cdot \varepsilon^\star_t\left( \sqrt{\bar{\alpha}_t} x_{\mathsf{noisy}} + \frac{\sqrt{\bar{u}_t} \sqrt{\bar{\alpha}_t} \eta^2 \widehat{Z}^{\mathsf{rev}}_{\tau^{\mathsf{rev}}_t}}{(\eta^2 - 1)\bar{u}_t + 1} \right),
\end{aligned}
$$

for $\tau^{\mathsf{rev}} \in [\tau_{t-1}, \tau_t]$, where

$$
h(\eta, u) := -\arctan \frac{\eta}{\sqrt{u^{-1} - 1}}.
$$

Plug in $\tau^{\text{rev}} = \tau_{t-1}$ in the above solution, and set $z_t = \widehat{Z}^{\text{rev}}_{\tau^{\text{rev}}_t}$, we obtain

$$z_{t-1} = \frac{\sqrt{(\eta^2 - 1)\bar{u}_{t-1} + 1}}{\sqrt{(\eta^2 - 1)\bar{u}_t + 1}} z_t \tag{40}$$
$$+ \sqrt{(\eta^2 - 1)\bar{u}_{t-1} + 1} \cdot \left(h(\eta, \bar{u}_{t-1}) - h(\eta, \bar{u}_t)\right) \cdot \varepsilon^\star_t \left(\sqrt{\bar{\alpha}_t} x_{\text{noisy}} + \frac{\sqrt{\bar{u}_t}\sqrt{\bar{\alpha}_t}\eta^2 z_t}{(\eta^2 - 1)\bar{u}_t + 1}\right).$$

The initialization, which should ideally be $z_{T'} = \widehat{Z}^{\text{rev}}_{\tau_\infty} \sim p_{Z_{\tau_\infty}}$, is approximated by $z_{T'} \sim \mathcal{N}(0, I_d)$. This is exactly the update equation and the initialization in Algorithm 3, except that $\varepsilon^\star_t$ is replaced by the noise estimate $\widehat{\varepsilon}_t$.

### E.4  Discretization of PCS

We first note that the Metropolis-adjustment step in PCS (cf. Algorithm 4) is standard following the classical form of MALA [RR98]. Therefore, we focus on explaining the Langevin step. Recall the continuous-time Langevin dynamics for sampling from the distribution $\exp(\mathcal{L}(\cdot; y) - \frac{1}{2\eta^2}\|\cdot - x\|^2)$:

$$\mathrm{d}Z_\tau = -\nabla_{Z_\tau}\mathcal{L}(Z_\tau; y)\mathrm{d}\tau + \frac{1}{\eta^2}(Z_\tau - x)\mathrm{d}\tau + \sqrt{2}\mathrm{d}B_\tau, \quad \tau \geq 0, \quad Z_0 \sim \mathcal{N}(0, I_d). \tag{41}$$

The classical form of MALA, as in [RR98], performs one step of a straightforward discretization of (41) as the Langevin step, as follows:

$$z_{n+\frac{1}{2}} \approx z_n - \gamma\nabla_{z_n}\mathcal{L}(z_n; y) + \frac{\gamma}{\eta^2}(z_n - x) + \sqrt{2\gamma}w_n, \quad w_n \sim \mathcal{N}(0, I_d).$$

In our setting, due to the presence of the linear drift term $\frac{1}{\eta^2}(Z_\tau - x)$, which can be quite large when $\eta$ is small, we apply the exponential integrator instead. Set the discretization time points $\tau_n = n\gamma$, the exponential integrator reads as

$$\mathrm{d}Z_\tau = -\nabla_{Z_{n\gamma}}\mathcal{L}(Z_{n\gamma}; y)\mathrm{d}\tau + \frac{1}{\eta^2}(Z_\tau - x)\mathrm{d}\tau + \sqrt{2}\mathrm{d}B_\tau, \quad n\gamma \leq \tau \leq (n+1)\gamma.$$

Solve this linear SDE on $n\gamma \leq \tau \leq (n+1)\gamma$ directly (see also (34)) to obtain

$$Z_{(n+1)\gamma} \overset{(\mathrm{d})}{=} rZ_{n\gamma} + (1 - r)x + \eta^2(1 - r)\nabla_{Z_{n\gamma}}\mathcal{L}(Z_{n\gamma}; y) + \eta\sqrt{1 - r^2}w_n, \quad w_n \sim \mathcal{N}(0, I_d),$$

where $r := \mathrm{e}^{-\gamma/\eta^2}$. This is the same as the update equation for the Langevin step in PCS (cf. Algorithm 4).

## F  Proof of main theorems

### F.1  Proof of Theorem 1

*Proof.* We first collect the asymptotic correctness of our subroutines PCS and DDS in the following two lemmas. The correctness of PCS is actually well-known, see e.g., [Tie94, Corollary 2].

**Lemma 3** (Correctness of PCS). *Under Assumption 1, with notation in Algorithm 4, in the continuous-time limit: $\gamma \to 0$, $N \to \infty$, the algorithm PCS outputs samples with distribution $\propto \exp(\mathcal{L}(\cdot; y) + \frac{1}{2\eta}\|\cdot - x\|^2)$.*

The next lemma guarantees the correctness of DDS with exact unconditional score functions.

**Lemma 4** (Correctness of DDS). *Assume the score function estimation $\widehat{s}_t$ is accurate, i.e. $\widehat{s}_t = s^\star_t$. In the continuous-time limit: $T \to \infty$, $\bar{\alpha}_T \to 0$, $\frac{\bar{\alpha}_{t-1}}{\bar{\alpha}_t} \to 1$, uniformly in $t$, both DDS-DDIM and DDS-DDPM output $x$ obeying the posterior distribution $p^\star(x^\star = x \mid x^\star + \eta\varepsilon = x_{\text{noisy}})$, $\varepsilon \sim \mathcal{N}(0, I_d)$.*

The proof of Theorem 1 is based on two lemmas on the one-step transition kernel of DPnP and the asymptotic behavior of the transition kernel, which we will present soon. First, we set up some notations. Denote

$$p_\eta(x) := p_{x^\star \sim p^\star, \varepsilon \sim \mathcal{N}(0, I_d)}(x^\star + \eta\varepsilon = x) = \frac{1}{(2\pi)^{d/2}\eta^d} \int p^\star(z)\mathrm{e}^{-\frac{1}{2\eta^2}\|x - z\|^2}\mathrm{d}z.$$

From the first equality, it is clear that $p_\eta \to p^\star$ when $\eta \to 0^+$. We will also use the notation $q_\eta$ defined in (24), which we recall here:

$$q_\eta(x) := \frac{1}{(2\pi)^{d/2}\eta^d} \int e^{\mathcal{L}(z;y) - \frac{1}{2\eta^2}\|x-z\|^2} \mathrm{d}z.$$

In virtue of the Assumption 1, we know that $q_\eta$ is finite for all $x \in \mathbb{R}^d$.

For convenience, we introduce a notation for application of transition kernels. For a probability distribution $p(x)$ and a probability transition kernel $K(x, x')$, denote by $p \circ K$ the probability distribution given by

$$p \circ K(x') = \int p(x)K(x, x')\mathrm{d}x.$$

The first lemma characterizes the one-step behavior of DPnP in terms of Markov transition kernels.

**Lemma 5.** *Under the settings of Lemma 4 and Lemma 3, the one-step transition kernel of DPnP with $\eta_k = \eta$ is given by:*

$$K_{\mathsf{DPnP},\eta}(x, x') = \left( \int \frac{q_0(z)}{p_\eta(z)} e^{-\frac{1}{2\eta^2}\|z-x\|^2 - \frac{1}{2\eta^2}\|z-x'\|^2} \mathrm{d}z \right) \frac{p^\star(x')}{q_\eta(x)}.$$

*In other words, if $\widehat{x}_k$ has distribution $p_{\widehat{x}_k}$, then the distribution of $\widehat{x}_{k+1}$ is*

$$p_{\widehat{x}_{k+1}}(x') = p_{\widehat{x}_k} \circ K_{\mathsf{DPnP},\eta}(x) = \int p_{\widehat{x}_k}(x)K_{\mathsf{DPnP},\eta}(x, x')\mathrm{d}x.$$

The proof is postponed to Appendix F.3. The next lemma analyzes the ergodic properties of the Markov chain with transition kernel $K_{\mathsf{DPnP},\eta}$. These properties are known [BB23] but scattered in different literatures, so we will provide a brief proof to be self-contained.

**Lemma 6.** *The Markov transition kernel $K_{\mathsf{DPnP},\eta}$ has the following properties:*

(i) *(Stationary distribution.) Let $\pi_\eta$ be the probability distribution defined by*

$$\pi_\eta(x) = c_\eta p^\star(x)q_\eta(x),$$

*where $c_\eta > 0$ is the normalization constant such that $\int \pi_\eta(x)\mathrm{d}x = 1$. Then $K_{\mathsf{DPnP},\eta}$ is reversible with stationary distribution $\pi_\eta$.*

(ii) *(Convergence.) For any initial distribution $p$, the distribution of the Markov chain with kernel $K_{\mathsf{DPnP},\eta}$ converges to $\pi_\eta$:*

$$\mathsf{TV}(p \circ K^{(n)}_{\mathsf{DPnP},\eta}, \pi_\eta) \to 0, \quad n \to \infty, \tag{42}$$

*where $K^{(n)}_{\mathsf{DPnP},\eta}$ is the $n$-step transition kernel of $K_{\mathsf{DPnP},\eta}$.*

The proof is postponed to Appendix F.4. We now show how to prove Theorem 1 with the above two lemmas. With the annealing schedule in Theorem 1, between steps $k_{l-1} \le k < k_l$, which consist of consecutive $(k_l - k_{l-1})$ steps, the transition kernel of one-step of DPnP is $K_{\mathsf{DPnP},\varepsilon_l}$. As $(k_l - k_{l-1}) \to \infty$, Lemma 6 implies that

$$\mathsf{TV}(p_{\widehat{x}_{k_l}}, \pi_{\varepsilon_l}) = \mathsf{TV}(p_{\widehat{x}_{k_{l-1}}} \circ K^{(k_l - k_{l-1})}_{\mathsf{DPnP},\varepsilon_l}, \pi_{\varepsilon_l}) \to 0.$$

Under the assumption in Theorem 1 that $\varepsilon_l \to 0$, we let $l \to \infty$ to see $\lim_{l \to \infty} \pi_{\varepsilon_l} = c_0 p^\star(\cdot)e^{\mathcal{L}(\cdot;y)} = p^\star(\cdot|y)$, thus $p_{\widehat{x}_{k_l}} \to p^\star(\cdot|y)$, as claimed. $\qquad\square$

### F.2 Proof of Lemma 4

*Proof.* For DDS-DDPM, we note that under the continuous-time limit in Lemma 4, the discretization time points given by (35) verify

$$\tau_0^{\mathsf{rev}} = 0, \quad \sup_{0 \le t \le T'-1} |\tau_t^{\mathsf{rev}} - \tau_{t+1}^{\mathsf{rev}}| \to 0, \quad \tau_{T'}^{\mathsf{rev}} \to \left(\frac{1}{1+\eta^2}\right)^{-1} - 1 = \eta^2, \quad T' \to \infty.$$

Therefore, these discretization time points $0 = \tau_0^{\mathsf{rev}} \le \cdots \le \tau_{T'}^{\mathsf{rev}} \le \eta^2$ form a partition of $[0, \eta^2]$, which becomes infinitely fine in the continuous-time limit. Thus the discretized integrator (37) converges to the solution of the SDE (16), which, as we have already argued in Appendix D, produces samples obeying the denoising posterior distribution $p^\star(\cdot|x_{\mathsf{noisy}})$, as claimed.

The proof for DDS-DDPM follows similarly, by observing that the discretization time points in (35) form an infinitely fine partition of $[0, \infty)$ in the continuous-time limit. $\qquad\square$

### F.3 Proof of Lemma 5

*Proof.* The proof is based on computing the transition kernel of the two subroutines. We claim that

(i) Sampling with probability density proportional to $\exp(\mathcal{L}(\cdot; y) - \frac{1}{2\eta^2}\|\cdot -x\|^2)$ is equivalent to applying the following Markov transition kernel

$$K_{\mathsf{PCS},\eta}(x, x') = \frac{1}{q_\eta(x)} e^{\mathcal{L}(x';y) - \frac{1}{2\eta^2}\|x'-x\|^2}.$$

(ii) Sampling with probability $p^\star(x^\star \mid x^\star + \eta\varepsilon = x)$, where $\varepsilon \sim \mathcal{N}(0, I_d)$, is equivalent to applying the following Markov transition kernel:

$$K_{\mathsf{DDS},\eta}(x, x') = \frac{1}{p_\eta(x)} p^\star(x') e^{-\frac{1}{2\eta^2}\|x'-x\|^2}.$$

It is then clear that

$$K_{\mathsf{DPnP},\eta}(x, x') = \int K_{\mathsf{PCS},\eta}(x, z) K_{\mathsf{DDS},\eta}(z, x') dz = \left(\int \frac{q_0(z)}{p_\eta(z)} e^{-\frac{1}{2\eta^2}\|z-x\|^2 - \frac{1}{2\eta^2}\|z-x'\|^2} dz\right) \frac{p^\star(x')}{q_\eta(x)},$$

as desired. We now prove the above two claims. For (i), note that by (23), we know $K_{\mathsf{DDS},\eta}(x, \cdot) \propto p^\star(\cdot) e^{-\frac{1}{2\eta^2}\|\cdot -x\|^2}$. Thus it suffices to compute the normalization constant, which is

$$\int p^\star(x') e^{-\frac{1}{2\eta^2}\|x'-x\|^2} dx' = p_\eta(x),$$

by the definition of $p_\eta$. Therefore

$$K_{\mathsf{DDS},\eta}(x, x') = \frac{1}{p_\eta(x)} p^\star(x') e^{-\frac{1}{2\eta^2}\|x'-x\|^2},$$

as claimed. The proof of (ii) follows similarly. $\qquad\square$

### F.4 Proof of Lemma 6

*Proof.* We first introduce a fundamental lemma [PW24], which provides a simple method to bound the total variation between two distributions.

**Lemma 7** (Data-processing inequality)**.** *Let $p, q$ be two probability distributions, and $K$ be a probability transition kernel. Then*

$$\mathsf{TV}(p \circ K, q \circ K) \le \mathsf{TV}(p, q).$$

We now prove the two items in Lemma 6 separately.

*Proof of (i).* We first show that $\pi_\eta$ is well-defined, i.e., $\int p^\star(x) q_n \eta(x) dx < \infty$. This can be seen from Assumption 1, which implies $q_\eta(x) \lesssim \int e^{-\frac{1}{2\eta^2}\|x-z\|^2} dz \lesssim 1$, hence

$$\int p^\star(x) q_\eta(x) dx \lesssim \int p^\star(x) dx = 1.$$

To show that $K_{\mathsf{DPnP},\eta}$ is reversible with stationary distribution $\pi_\eta$, it suffices to verify
$$\pi_\eta(x)K_{\mathsf{DPnP},\eta}(x,x') = \pi_\eta(x')K_{\mathsf{DPnP},\eta}(x',x), \quad \forall x, x' \in \mathbb{R}^d.$$
However, it is easily checked that both sides are equal to
$$c_\eta \left( \int \frac{q_0(z)}{p_\eta(z)} e^{-\frac{1}{2\eta^2}\|z-x\|^2 - \frac{1}{2\eta^2}\|z-x'\|^2} dz \right) p^\star(x') p^\star(x).$$

*Proof of (ii).* We define an auxiliary Markov transition kernel $K_{\mathsf{aux},\eta} = K_{\mathsf{DDS},\eta} \circ K_{\mathsf{PCS},\eta}$. More explicitly,
$$K_{\mathsf{aux},\eta}(x,x') = \int K_{\mathsf{DDS},\eta}(x,z) K_{\mathsf{PCS},\eta}(z,x') dz = \left( \int \frac{p^\star(z)}{q_\eta(z)} e^{-\frac{1}{2\eta^2}\|z-x\|^2 - \frac{1}{2\eta^2}\|z-x'\|^2} dz \right) \frac{e^{\mathcal{L}(x';y)}}{p_\eta(x)}. \tag{43}$$

It is easy to see that
$$p \circ K_{\mathsf{DPnP},\eta}^{(n)} = p \circ K_{\mathsf{PCS},\eta} \circ K_{\mathsf{aux},\eta}^{(n-1)} \circ K_{\mathsf{DDS},\eta}. \tag{44}$$
Thus we are led to investigate the ergodic properties of $K_{\mathsf{aux},\eta}$. Similar to the proof of item (i) above, it is not hard to show that $K_{\mathsf{aux},\eta}$ is reversible with respect to the stationary distribution
$$\mu_\eta(x) := c_\eta p_\eta(x) q_0(x) = c_\eta p_\eta(x) e^{\mathcal{L}(x;y)}.$$
Moreover, one may check that
$$\pi_\eta = \mu_\eta \circ K_{\mathsf{DDS},\eta}. \tag{45}$$
It is apparent that $\mu(x') > 0$ and $K_{\mathsf{aux},\eta}(x,x')/\mu_\eta(x') > 0$ for all $x, x' \in \mathbb{R}^d$. By [Tie94, Corollary 1], such a Markov transition kernel obeys, for any probability distribution $q$, that
$$\mathsf{TV}(q \circ K_{\mathsf{aux},\eta}^{(n)}, \mu_\eta) \to 0, \quad n \to \infty.$$
In view of (44) and (45), we set $q = p \circ K_{\mathsf{PCS},\eta}$ and invoke the data-processing inequality to obtain
$$\begin{aligned}
\mathsf{TV}(p \circ K_{\mathsf{DPnP},\eta}^{(n)}, \pi_\eta) &= \mathsf{TV}(q \circ K_{\mathsf{aux},\eta}^{(n-1)} \circ K_{\mathsf{DDS},\eta}, \mu_\eta \circ K_{\mathsf{DDS},\eta}) \\
&\le \mathsf{TV}(q \circ K_{\mathsf{aux},\eta}^{(n-1)}, \mu_\eta) \\
&\to 0,
\end{aligned}$$
as $n \to \infty$. This completes the proof. $\qquad\square$

### F.5 Proof of Theorem 2

*Proof.* Denote by $\tilde{K}_{\mathsf{PCS},\eta}$ and $\tilde{K}_{\mathsf{DDS},\eta}$ and the transition kernels for $\mathsf{PCS}$ and for $\mathsf{DDS}$, respectively. Note that these may deviate from the transition kernels $K_{\mathsf{PCS},\eta}$ and $K_{\mathsf{DDS},\eta}$ defined for the idealized asymptotic setting in Appendix F. We have
$$\begin{aligned}
\mathsf{TV}(p_{\widehat{x}_N}, \pi_\eta) &= \mathsf{TV}(p_{\widehat{x}_{N-\frac{1}{2}}} \circ \tilde{K}_{\mathsf{DDS},\eta}, \pi_\eta) \\
&\le \mathsf{TV}(p_{\widehat{x}_{N-\frac{1}{2}}} \circ K_{\mathsf{DDS},\eta}, \pi_\eta) + \mathsf{TV}(p_{\widehat{x}_{N-\frac{1}{2}}} \circ K_{\mathsf{DDS},\eta}, p_{\widehat{x}_{N-\frac{1}{2}}} \circ \tilde{K}_{\mathsf{DDS},\eta}) \\
&\le \mathsf{TV}(p_{\widehat{x}_{N-\frac{1}{2}}} \circ K_{\mathsf{DDS},\eta}, \pi_\eta) + \varepsilon_{\mathsf{DDS}},
\end{aligned}$$

where the second line is triangle inequality, and the third line follows from the assumption in Theorem 2 that $\mathsf{DDS}$ has error at most $\varepsilon_{\mathsf{DDS}}$ in total variation, by taking the input of $\mathsf{DDS}$ to be $\widehat{x}_{N-\frac{1}{2}}$.

Similarly, from $p_{\widehat{x}_{N-\frac{1}{2}}} = p_{\widehat{x}_{N-1}} \circ \tilde{K}_{\mathsf{PCS},\eta}$ and the assumption that $\mathsf{PCS}$ has error at most $\varepsilon_{\mathsf{PCS}}$ in total variation, we can show
$$\mathsf{TV}(p_{\widehat{x}_{N-\frac{1}{2}}} \circ K_{\mathsf{DDS},\eta}, \pi_\eta) \le \mathsf{TV}(p_{\widehat{x}_{N-1}} \circ K_{\mathsf{PCS},\eta} \circ K_{\mathsf{DDS},\eta}, \pi_\eta) + \varepsilon_{\mathsf{PCS}} = \mathsf{TV}(p_{\widehat{x}_{N-1}} \circ K_{\mathsf{DPnP},\eta}, \pi_\eta) + \varepsilon_{\mathsf{PCS}}.$$

The above two inequalities together imply
$$\mathsf{TV}(p_{\widehat{x}_N}, \pi_\eta) \le \mathsf{TV}(p_{\widehat{x}_{N-1}} \circ K_{\mathsf{DPnP},\eta}, \pi_\eta) + \varepsilon_{\mathsf{DDS}} + \varepsilon_{\mathsf{PCS}}.$$
Iterating this process, we obtain
$$\mathsf{TV}(p_{\widehat{x}_N}, \pi_\eta) \le \mathsf{TV}(p_{\widehat{x}_1} \circ K_{\mathsf{DPnP},\eta}^{(N-1)}, \pi_\eta) + (N-1)(\varepsilon_{\mathsf{DDS}} + \varepsilon_{\mathsf{PCS}}). \tag{46}$$

It remains to bound $\mathsf{TV}(p_{\widehat{x}_1} \circ K_{\mathsf{DPnP},\eta}^{(N-1)}, \pi_\eta)$. For this, we need the following two lemmas.

**Lemma 8** (Comparing TV and $\chi^2$-divergence, [PW24]). *For any two distributions $p, q$, we have*

$$\mathsf{TV}(p, q) \le \sqrt{\chi^2(p \,\|\, q)}.$$

**Lemma 9** ($\chi^2$-contractivity of $K_{\mathsf{DPnP},\eta}$). *There exists some $\lambda := \lambda(p^\star, \mathcal{L}, \eta) \in (0, 1)$, such that for any probability distribution $p(x)$, we have*

$$\chi^2(p \circ K_{\mathsf{DPnP},\eta}^{(N)} \,\|\, \pi_\eta) \le \lambda^{2N} \chi^2(p \,\|\, \pi_\eta).$$

A form of Lemma 9 is well-known for Markov chains with countable state spaces, but relatively few sources provide a complete proof for the abstract setting we consider here with continuous state space. For sake of completeness, we prove Lemma 9 in Appendix F.6.

Combining the above two lemmas, we obtain

$$\mathsf{TV}(p_{\widehat{x}_1} \circ K_{\mathsf{DPnP},\eta}^{(N-1)}, \ \pi_\eta) \le \sqrt{\chi^2(p_{\widehat{x}_1} \circ K_{\mathsf{DPnP},\eta}^{(N-1)} \,\|\, \pi_\eta)} \le \lambda^{N-1} \sqrt{\chi^2(p_{\widehat{x}_1} \,\|\, \pi_\eta)}.$$

Plug this into (46), we obtain

$$\mathsf{TV}(p_{\widehat{x}_N}, \pi_\eta) \le \lambda^{N-1} \sqrt{\chi^2(p_{\widehat{x}_1} \,\|\, \pi_\eta)} + (N-1)(\varepsilon_{\mathsf{DDS}} + \varepsilon_{\mathsf{PCS}}).$$

With $N \asymp \frac{\log(1/\varepsilon_{\mathsf{acc}})}{1-\lambda}$ such that $\lambda^{N-1} \le \exp\big(-(N-1)(1-\lambda)\big) \le \varepsilon_{\mathsf{acc}}$, the desired result readily follows. $\qquad\square$

### F.6 Proof of Lemma 9

*Proof.* We need a few fundamental properties of reversible Markov chains, which are collected below.

First we set up some notations. Define the Hilbert space $L^2(\pi)$ to be the space of square-integrable functions with respect to measure $\pi$, i.e., those functions $f : \mathbb{R}^d \to \mathbb{C}$ such that

$$\|f\|_{L^2(\pi)} := \left(\int |f(x)|^2 \pi(x)\mathrm{d}x\right)^{1/2} < \infty.$$

The first well-known property [SC97] offers a way to represent a reversible transition kernel as a self-adjoint operator (infinite-dimensional symmetric matrix).

**Lemma 10** (Self-adjoint representation of reversible Markov operator). *Assume $K(x, x')$ is a Markov transition kernel that is reversible with respect to the stationary distribution $\pi(x)$. Then the integral operator $\mathcal{K} : L^2(\pi) \to L^2(\pi)$ defined by*

$$\mathcal{K}f(x) = \int K(x, x')f(x')\mathrm{d}x'$$

*is self-adjoint and compact. For any probability distribution $p(x)$ such that $\int \frac{p^2(x)}{\pi(x)}\mathrm{d}x < \infty$, we have*

$$\int p(x) \cdot \mathcal{K}f(x)\mathrm{d}x = \int p \circ K(x')f(x')\mathrm{d}x'.$$

*Moreover, the eigenvalues of $\mathcal{K}$ are the same as those of $K$.*

The following theorem is a generalization of the classical Perron-Frobenius theory for finite-dimensional transition matrix to strictly positive operators. The form we present here can be found in [Sch12, Theorem V.6.6]; see also [Bou23b, Theorem III.6.7] for a more elementary treatment which can also be adapted to the form we need.

**Theorem 3** (Jentzsch). *Let $K(x, x')$ be a Markov transition kernel. If $K(x, x') > 0$ for any $x, x' \in \mathbb{R}^d$, then $K$ has a unique stationary distribution $\pi$. Moreover, $1$ is a simple eigenvalue of $K$, with $\pi$ being the only left eigenfunction, and the constant function $1$ being the only right eigenfunction. In addition, there exists $\lambda \in (0, 1)$ such that any other eigenvalue of $K$ has modulus no larger than $\lambda$.*

We are now ready to prove Lemma 9. We divide the proof into the following steps.

**Step 1: controlling the eigenvalues of $\mathcal{K}_{\mathsf{DPnP},\eta}$.** Recall the auxiliary kernel $K_{\mathsf{aux},\eta}$ defined in (43). It is a standard result in linear algebra or function analysis [Bou23a] that $K_{\mathsf{aux},\eta} = K_{\mathsf{PCS},\eta} \circ K_{\mathsf{DDS},\eta}$ has same eigenvalues as $K_{\mathsf{DPnP},\eta} = K_{\mathsf{DDS},\eta} \circ K_{\mathsf{PCS},\eta}$. From (43), it is easy to check $K_{\mathsf{aux},\eta}(x,x') > 0$, thus Theorem 3 implies 1 is a simple eigenvalue of $\mathcal{K}_{\mathsf{DPnP},\eta}$. Moreover, there exists $\lambda := \lambda(p^\star, \mathcal{L}, \eta) \in (0,1)$, such that any other eigenvalue of $K_{\mathsf{aux},\eta}$ has modulus no larger than $\lambda$.

Since $K_{\mathsf{DPnP},\eta}$ has the same eigenvalues as $K_{\mathsf{aux},\eta}$, and, by Lemma 10, the operator $\mathcal{K}_{\mathsf{DPnP},\eta}$ also has the same eigenvalues as these two, we conclude that $\mathcal{K}_{\mathsf{DPnP},\eta}$ is a self-adjoint compact operator on $L^2(\pi_\eta)$, of whom 1 is a simple eigenvalue. Moreover, any other eigenvalue of $\mathcal{K}_{\mathsf{DPnP},\eta}$ has modulus no larger than $\lambda$.

**Step 2: establishing the contractivity of $\mathcal{K}_{\mathsf{DPnP},\eta}$ in $L^2(\pi_\eta)$.** It is easy to verify that the constant function $\mathbf{1}$, which takes value 1 for any $x \in \mathbb{R}^d$, is a eigenfunction of $\mathcal{K}_{\mathsf{DPnP},\eta}$ associated to the simple eigenvalue 1, thus is the only (up to scaling) eigenfunction associated to that eigenvalue. It is also a unit-length eigenfunction, since $\|\mathbf{1}\|_{L^2(\pi_\eta)} = (\int 1 \cdot \pi_\eta(x)dx)^{1/2} = 1$. Therefore, the operator $\mathcal{K}_{\mathsf{DPnP},\eta} - \mathbf{1}\mathbf{1}^\top$ is a self-adjoint operator whose eigenvalues have modulus no larger than $\lambda$, where $\mathbf{1}\mathbf{1}^\top$ is the orthogonal projection onto $\mathbf{1}$ in $L^2(\pi_\eta)$, defined by

$$\mathbf{1}\mathbf{1}^\top f(x) \equiv \int f(x')\pi_\eta(x')\mathrm{d}x', \quad \forall x \in \mathbb{R}^d.$$

Using the fact that $\mathcal{K}_{\mathsf{DPnP},\eta}\mathbf{1}\mathbf{1}^\top = \mathbf{1}\mathbf{1}^\top\mathcal{K}_{\mathsf{DPnP},\eta} = \mathbf{1}\mathbf{1}^\top$, one may show $(\mathcal{K}_{\mathsf{DPnP},\eta} - \mathbf{1}\mathbf{1}^\top)^N = \mathcal{K}_{\mathsf{DPnP},\eta}^{(N)} - \mathbf{1}\mathbf{1}^\top$ by expanding the product, see e.g. [SC97]. Consequently, $\mathcal{K}_{\mathsf{DPnP},\eta}^N - \mathbf{1}\mathbf{1}^\top$ is a self-adjoint operator whose eigenvalues have modulus no larger than $\lambda^N$, i.e.,

$$\left\|\mathcal{K}_{\mathsf{DPnP},\eta}^N - \mathbf{1}\mathbf{1}^\top\right\|_{L^2(\pi_\eta) \to L^2(\pi_\eta)} \leq \lambda^N, \tag{47}$$

where $\|\cdot\|_{L^2(\pi_\eta) \to L^2(\pi_\eta)}$ denotes the operator norm on $L^2(\pi_\eta)$.

**Step 3: bounding the inner product of $p \circ K_{\mathsf{DPnP},\eta}^{(N)} - \pi_\eta$ with any square-integrable function.** Note that when $\chi^2(p\,\|\,\pi_\eta) = \infty$, the conclusion is trivially true. For the rest part of the proof, we assume $\chi^2(p\,\|\,\pi_\eta) < \infty$. Now, for any $f \in L^2(\pi_\eta)$, by applying Lemma 10 iteratively, we obtain

$$\int p \circ K_{\mathsf{DPnP},\eta}^{(N)}(x)f(x)\mathrm{d}x = \int p(x')\mathcal{K}_{\mathsf{DPnP},\eta}^N f(x')\mathrm{d}x'$$
$$= \int p(x') \cdot (\mathcal{K}_{\mathsf{DPnP},\eta}^N - \mathbf{1}\mathbf{1}^\top)f(x')\mathrm{d}x' + \int p(x')\mathbf{1}\mathbf{1}^\top f(x')\mathrm{d}x'$$
$$= \int p(x') \cdot (\mathcal{K}_{\mathsf{DPnP},\eta}^N - \mathbf{1}\mathbf{1}^\top)f(x')\mathrm{d}x' + \int f(x')\pi_\eta(x')\mathrm{d}x',$$

where the last line follows from the definition of $\mathbf{1}\mathbf{1}^\top$ and $\int p(x')\mathrm{d}x' = 1$. Rearrange the terms to see

$$\int \left(p \circ K_{\mathsf{DPnP},\eta}^{(N)}(x) - \pi_\eta(x)\right)f(x)\mathrm{d}x = \int p(x') \cdot (\mathcal{K}_{\mathsf{DPnP},\eta}^N - \mathbf{1}\mathbf{1}^\top)f(x')\mathrm{d}x'. \tag{48}$$

In particular, taking $p = \pi_\eta$ yields

$$0 = \int \pi_\eta(x') \cdot (\mathcal{K}_{\mathsf{DPnP},\eta}^N - \mathbf{1}\mathbf{1}^\top)f(x')\mathrm{d}x'. \tag{49}$$

Substract (49) from (48), and then take absolute value, we obtain

$$\left|\int \left(p \circ K_{\mathsf{DPnP},\eta}^{(N)}(x) - \pi_\eta(x)\right)f(x)\mathrm{d}x\right|$$
$$= \left|\int (p(x') - \pi_\eta(x')) \cdot (\mathcal{K}_{\mathsf{DPnP},\eta}^N - \mathbf{1}\mathbf{1}^\top)f(x')\mathrm{d}x'\right|$$
$$\leq \left(\int \frac{(p(x') - \pi_\eta(x'))^2}{\pi_\eta(x)}\mathrm{d}x\right)^{1/2} \cdot \left\|(\mathcal{K}_{\mathsf{DPnP},\eta}^N - \mathbf{1}\mathbf{1}^\top)f(x')\right\|_{L^2(\pi_\eta)}$$

$$\leq \sqrt{\chi^2(p \,\|\, \pi_\eta)} \cdot \lambda^N \|f\|_{L^2(\pi_\eta)}. \tag{50}$$

**Step 4: choosing an appropriate square-integrable function.** Now, set

$$f(x) = \frac{p \circ K_{\mathsf{DPnP},\eta}^{(N)}(x) - \pi_\eta(x)}{\pi_\eta(x)}.$$

It is easily checked that

$$\int \left( p \circ K_{\mathsf{DPnP},\eta}^{(N)}(x) - \pi_\eta(x) \right) f(x)\mathrm{d}x = \chi^2(p \circ K_{\mathsf{DPnP},\eta}^{(N)} \,\|\, \pi_\eta),$$

$$\|f\|_{L^2(\pi_\eta)} = \sqrt{\chi^2(p \circ K_{\mathsf{DPnP},\eta}^{(N)} \,\|\, \pi_\eta)}.$$

Plug these equations into (50), we obtain

$$\chi^2(p \circ K_{\mathsf{DPnP},\eta}^{(N)} \,\|\, \pi_\eta) \leq \lambda^{2N} \chi^2(p \,\|\, \pi_\eta),$$

as claimed. □

# G   Additional numerical results

## G.1   Implementation details

*Score functions used.* We use the same pre-trained score functions as in [CKM+23].[3]

*Normalization.* All images are normalized in the usual way to fit into the range $[-1, 1]$.

*Parameters of our algorithm.* We choose the same annealing schedule across all tasks. Please see Appendix H for a detailed discussion.

*Parameters of comparison methods.* We made our best effort to fine-tune the other algorithms within a reasonable amount of time for each task. We list the paramters, following the notations in the original paper [CKM+23, SZY+23], as follows.

- Super-resolution: For DPS, the learning rate is set to $0.6$. For LGD-MC, the MC sampling variance $r_t = 0.05$, the loss coefficient $\lambda = 10^{-3}$, and the learning rate is set to $60.0$. For ReSample, the stochastic resampling variance parameter $\gamma = 10$ for FFHQ and $\gamma = 4.0$ for ImageNet.

- Phase retrieval: For DPS, the stepsize is set to $0.8$. For LGD-MC, the MC sampling variance $r_t = 0.05$, the loss coefficient $\lambda = 10^{-3}$, and the learning rate is set to $400.0$.

- Quantized sensing: For DPS, the stepsize is set to $100.0$. For LGD-MC, the MC sampling variance $r_t = 0.05$, the loss coefficient $\lambda = 2 \times 10^{-5}$, and the learning rate is set to $500.0$. For ReSample, the stochastic resampling variance parameter $\gamma = 4$ for FFHQ and $\gamma = 3.5$ for ImageNet.

## G.2   Forward measurement operators

*Super-resolution.* The forward model for super-resolution is the usual bicubic downsampling operator [Key81], which is a linear operator (in fact, a block Hankel matrix). We use a downsampling ratio of $4$ in all our experiments. The measurement noise is set to be white Gaussian, with variance $0.2$. Note that the noise variance is moderately larger than that in [CKM+23] to better reflect the scenario in practical inverse problems.

*Phase retrieval.* We consider phase retrieval with a coded mask, which is a classical inverse problem [CLS15]. For a $256 \times 256$ image $x$ (for each color channel) in our experiments, we first generate a random mask $M \in \mathbb{R}^{256 \times 256}$ (which is shared across color channels), then apply Fourier transform $\mathcal{F}$ to $M \odot x$, where $\odot$ denotes the Hadarmard (entrywise) product, and finally preserve only the magnitudes of the Fourier transform. Formally, the forward measurement operator is $\mathcal{A}(x) = \mathrm{mag}(\mathcal{F}(M \odot x))$, where $\mathrm{mag}(\cdot)$ computes the entrywise magnitude of a matrix with complex entries. The measurement noise is again set to be white Gaussian, with variance $0.2$.

---

[3]https://github.com/DPS2022/diffusion-posterior-sampling

Table 5: Samples of different algorithms for super-resolution (4x).

| Input | DPS | LGD-MC | ReSample | DPnP-DDPM | DPnP-DDIM | Ground truth |
|---|---|---|---|---|---|---|

*Quantized sensing.* In this work, quantized sensing refers to the task of reconstructing an image from its low-bit quantized version. The forward measurement operator is a one-bit per channel, dithered quantization operator. More precisely, the forward measurement operator is an entrywise application of the following stochastic function $Q$ with dithering level $\theta > 0$:

$$Q(\text{pixel}) = \begin{cases} 1, & \text{with probability } \frac{e^{\text{pixel}/\theta}}{1+e^{\text{pixel}/\theta}} \\ -1, & \text{with probability } \frac{1}{1+e^{\text{pixel}/\theta}}, \end{cases}$$

where pixel $\in [-1, 1]$ is the value of each pixel in each channel. The measurements in quantized sensing are therefore one-bit-per-channel images. The dithering level $\theta$ is set to $0.4$ in our experiments.

### G.3 Sample images for other inverse problems

*Super-resolution.* The samples generated by different algorithms are shown in Table 5.

Across different tasks, linear and nonlinear, it can be seen that DPnP has stronger capability of reconstructing the image with higher fidelity to the fine details.

### G.4 Additional performance metrics

*Computation time in terms of Neural Function Estimations (NFEs).* In additional to the clock time statistics in the main text, we also measure the computational cost per sample of different algorithms in terms of the number of Neural Function Estimations, i.e., the number of calls to score functions. The results are in Table 6. Note that the NFEs for DPnP depends on the initialization, the annealing schedule (and the number of timesteps for DPnP-DDIM). We provide typical numbers of NFEs with the choice of parameters given in Appendix H and with a suitable number of timesteps.

Table 6: Number of NFEs for different algorithms.

| Algorithm | DPnP-DDIM | DPnP-DDPM | DPS | LGD-MC | ReSample |
|---|---|---|---|---|---|
| NFEs | $\sim 1500$ | $\sim 3000$ | 1000 | 1000 | 1000 |

*Frechet Inception Distance (FID) and Structural Similarity Index Measure (SSIM).* We also compare the FID and SSIM of different algorithms across different tasks. The results are shown in Table 7 and Table 8. It should be pointed out that FID is arguably not a very relevant notion to measure the quality of solving inverse problems, as accurately solving inverse problems means that the generated distribution is close to the *conditional*, i.e., *posterior* distribution of the image, while FID only measure the closeness to the *unconditional*, i.e., *prior* distribution of the image.

Table 7: FID and SSIM of solving inverse problems on FFHQ $256 \times 256$ validation dataset (1k samples).

| | Super-resolution (4x, linear) | | Phase retrieval (nonlinear) | | Quantized sensing (nonlinear) | |
|---|---|---|---|---|---|---|
| Algorithm | FID $\downarrow$ | SSIM $\uparrow$ | FID $\downarrow$ | SSIM $\uparrow$ | FID $\downarrow$ | SSIM $\uparrow$ |
| DPnP-DDIM (ours) | **36.3** | **0.668** | **46.5** | **0.631** | **37.3** | **0.712** |
| DPS [CKM$^+$23] | 38.6 | 0.636 | 52.0 | 0.494 | 42.1 | 0.601 |
| LGD-MC ($n = 5$) [SZY$^+$23] | 36.8 | 0.651 | 82.3 | 0.414 | 40.3 | 0.639 |
| ReSample (pixel-based) [SKZ$^+$23] | 40.2 | 0.641 | - | - | 40.0 | 0.657 |

Table 8: FID and SSIM of solving inverse problems on ImageNet $256 \times 256$ validation dataset (1k samples).

| | Super-resolution (4x, linear) | | Phase retrieval (nonlinear) | | Quantized sensing (nonlinear) | |
|---|---|---|---|---|---|---|
| Algorithm | FID $\downarrow$ | SSIM $\uparrow$ | FID $\downarrow$ | SSIM $\uparrow$ | FID $\downarrow$ | SSIM $\uparrow$ |
| DPnP-DDIM (ours) | 47.5 | **0.510** | **73.5** | 0.289 | **43.2** | **0.623** |
| DPS [CKM$^+$23] | 61.4 | 0.496 | 92.7 | **0.318** | 82.4 | 0.459 |
| LGD-MC ($n = 5$) [SZY$^+$23] | **46.2** | 0.503 | 89.8 | 0.234 | 46.8 | 0.563 |
| ReSample (pixel-based) [SKZ$^+$23] | 68.3 | 0.427 | - | - | 53.7 | 0.491 |

# H  Ablation studies

## H.1  Initialization

In the Algorithm 1, the initial guess $\widehat{x}_0$ is set to be a properly scaled Gaussian random vector. However, from Theorem 2 it can be infered that using a heuristic posterior sampler as the initializer could decrease $\chi^2(p_{\widehat{x}_1} \| \pi_\eta)$, hence potentially improve the convergence speed of DPnP. By using existing algorithms like DPS or LGD-MC as initializer, DPnP can improve upon the results of existing algorithms towards the correct posterior distribution efficiently and provably. In our experiments, we find it helpful to initialize DPnP with LGD-MC, which accelerates the algorithm significantly.

## H.2  Annealing schedule

We discuss the choice of the annealing schedule $\eta_k$ in DPnP (Algorithm 1). As seen in the theoretical analysis (Theorem 2), if we set all the $\eta_k \equiv \eta$ for some constant $\eta > 0$, then DPnP converges to a distribution $\pi_\eta$, which can be regarded as a version of the posterior distribution $p^\star(\cdot|y)$ distorted by an order of $O(\eta)$. The smaller $\eta$ is, the more accurate the final distribution will be. On the other hand, it was also seen that in many cases, the spectral gap is $\Omega(\eta)$, hence the convergence time is $O(\frac{1}{\eta})$.

Therefore, smaller $\eta$ would make it take longer to converge.[4]

To strike a balance between the accuracy and the convergence rate, we find it empirically successful to adapt an gradually decreasing schedule for $\eta_k$, similar to [BB23]. In the first few iterations, we set $\eta_k$ to be a large constant. After this initial phase, we decrease $\eta_k$ slowly, eventually to $\eta_N$ which is chosen to be a small constant. An example of such an annealing schedule is

$$\eta_0 = \eta_1 = \cdots = \eta_{K_0}, \quad \eta_0 > 0 \text{ is a large constant,}$$

$$\eta_k = (\eta_K/\eta_0)^{\frac{k - K_0}{K - K_0}} \eta_0, \quad K_0 < k \leq K, \quad \eta_K > 0 \text{ a small constant,}$$

where $K_0 < K$ is the length of the initial phase, which can be chosen as, e.g. $K_0 = K/5$. For all the numerical experiments, we set $\eta_0 = 0.4$, $\eta_N = 0.15$, $K_0 = 4$, $K = 20$.

---

[4]Strictly speaking, while the number of iterations required to converge increases as $\eta$ gets smaller, the computational complexity per iteration will decrease. However, in experiments, we observed that the latter effect is not strong enough to compensate for the increase in overall complexity caused by the former.

