# OpenReview forum: "Provably Robust Score-Based Diffusion Posterior Sampling for Plug-and-Play Image Reconstruction"
_NeurIPS.cc/2024/Conference — NeurIPS 2024 poster_

### Official Review · Reviewer_juip · 2024-07-01

**Soundness:** 4
**Presentation:** 3
**Contribution:** 3
**Rating:** 7
**Confidence:** 3

**Summary:**

The authors propose 'diffusion plug-and-play' (DPnP), a plug-and-play diffusion framework that alternatively calls what amounts to a consistency sampler based on the likelihood of the forward model, followed by (essentially) and unconditional diffusion step using the score function. The authors provide many subsequent theoretical results for this framework.

**Strengths:**

1. The paper is well-written and easy to follow/understand (barring some notation issues, see Weaknesses).
2. Each step in the process of developing the DPnP algorithm is well-motivated and explained thoroughly.
3. The provided proofs for the main theorems in Appendix F are well-written and seem correct.
4. DPnP outperforms competitors in non-linear inverse problems in nearly all tested metrics in Section 5 and Appendix G.
5. The overall contribution is impactful with respect to non-linear inverse problems.

**Weaknesses:**

1. This paper would benefit from improvements to the mathematical notation. Namely, it would be easier to read the math if scalar quantities were better differentiated from vector quantities (e.g., bold-faced vectors).
2. The formatting structure of the main paper is, at times, very granular with many lists. Personally, I appreciate this, but I feel that others may think that things could be ordered/structured better. This is a very minor issue, but worth keeping in mind.
3. I acknowledge that the primary contribution of this paper is the theoretical results associated with the DPnP framework, but I think that more robust experimental evaluation would have benefited this work. In particular, it would have been nice to see how the DPnP framework performs on more 'typical' linear inverse problems (e.g., inpainting, deblurring). This would have given a better sense of the overall performance of the approach, even if the focus is performance on non-linear inverse problems. I do not expect you to perform these experiments in the revision/for the rebuttal, I just wanted to note that the results would be more convincing if there were more of them.
4. It would be nice to discuss the connections between the proximal consistency sampler and projection-type approaches (e.g., that leveraged by Chung et al.'s 'Come-Closer-Diffuse-Faster) when the inverse problem is linear. In fact, I suspect that, for linear inverse problems, DPnP can be reduced to a simple projection step, followed by the reverse diffusion step. It would be worthwhile to discuss these connections, even if the discussion is relegated to an appendix.
5. In Algorithm 1, proximal consistency sampling is done before the denoising diffusion sampling. Why is this? The previously mentioned projection methods have the steps flipped, so I wonder why you have decided to structure the DPnP algorithm this way.

**Questions:**

See Weaknesses.

**Limitations:**

The authors adequately discussed the limitations of their method.

---

> ### Author Rebuttal · Authors · 2024-08-06
>
> Thank you for your positive feedback and insightful comments. We will make sure to take your invaluable suggestions into account in revising the paper and in our future work. Below we address the two questions you raise.
>
> **Connections with projection-type approaches.**
> - Thank you for your thoughtful question. We are willing to clarify our connection and difference with projection-type approaches as follows.
> > While our algorithm draws inspiration from projection-type approaches, the essential difference between our algorithm and these approaches persists even for linear problems. Roughly speaking, our proximal consistency step can be reduced (cf. Line 274-277 in our paper) to a slightly more complicated, yet theoretically rigorous version of the (proximal) projection step in these approaches for linear problems, but the same _cannot_ be said about our denoising diffusion step.
> >
> > A manifestation of such essential difference is that our denoising diffusion step requires multiple steps of reverse diffusion in a single iteration of outer loop, and has an accurate, simple expression for the (stationary) distribution it aims to sample from. In contrast, for projection-type approaches, it is typical that only one step of reverse diffusion is run in a single iteration of outer loop, and the distribution of its output, even in the ideal continuous setting, cannot be expressed in a simple way, which significantly hinders theoretical analysis. From a big picture, by running more steps of reverse diffusion every time after we do proximal consistency sampling, we stabilize the process since we better compensate for the drifts from the data manifold induced by (proximal) projection onto the null space of consistency equations. This is the heuristic reason for the improved robustness of our algorithm, which is backed by rigorous mathematical proofs.
>
>
> **Order of proximal consistency step and denoising diffusion step.**
> - Thank you for your sharp observation. The first reason for such an order is that the algorithm would have visually better outputs if ending with the denoising diffusion step instead of the proximal consistency step, as denoising diffusion by design outputs visually satisfactory images while proximal consistency sampler only cares about consistency. Such a difference was not present in previous projection-type approaches, as the denoising stage there runs only one step of reverse diffusion, thus it is insignificant whether it comes as the penultimate step or the last step. This again constitutes an example of the difference of our algorithm. The second reason is pedagogical, since this order better parallels that of projected gradient descent, which may facilitate the understanding of the logic of our algorithm.

---

> > ### Comment · Reviewer_juip · 2024-08-11
> >
> > Thank you for addressing my concerns. I think that this work has value and I will keep my original score of 7.

---

### Official Review · Reviewer_ppwi · 2024-07-04

**Soundness:** 3
**Presentation:** 3
**Contribution:** 2
**Rating:** 5
**Confidence:** 4

**Summary:**

This paper introduces a diffusion plug-and-play method (DPnP) that uses score-based diffusion models as expressive data priors for nonlinear inverse problems with general forward models. By combining a proximal consistency sampler and a denoising diffusion sampler, the method offers provably robust posterior sampling, with performance guarantees and demonstrated effectiveness across various tasks.

**Strengths:**

This paper establishes both asymptotic and non-asymptotic performance guarantees for DPnP and provides numerical experiments to demonstrate its effectiveness across various tasks. The theoretical analysis presented is a valuable contribution to the field.

**Weaknesses:**

Although I appreciate the theoretical aspect of this paper, as mentioned in the abstract, “this paper develops an algorithmic framework for employing score-based diffusion models as an expressive data prior in nonlinear inverse problems.” However, there are already existing works on embedding denoising diffusion models into plug-and-play frameworks as data priors, such as [1-2]. It would be helpful if the author could clearly highlight any new insights within this paper to distinguish it from previous work; otherwise, the novelty of this paper may appear somewhat incremental.

Minor one: There are several typos, e.g., in the abstract, the sentence "Score-based diffusion models, thanks to its impressive empirical success, have emerged as an appealing candidate of an expressive prior in image reconstruction." The correct pronoun should be "their" instead of "its" to match the plural subject "Score-based diffusion models."

**Questions:**

I wonder if the asymptotic consistency and non-asymptotic error analysis of DPnP, as established in this paper, demonstrate convergence. If so, it is necessary to verify this through numerical experiments.

Could you please explain why the result of DPnPDDPM shown in Table 1 is smooth, while the one shown in Table 5 is noisy with a lot of noticeable noise? Thank you.

**Limitations:**

As noted in the weaknesses section, the primary limitation is that diffusion-based PnP methods have already been proposed in [1-2] and applied to various inverse imaging problems.

---

> ### Author Rebuttal · Authors · 2024-08-04
>
> Thank you for your positive feedback on our theoretical contribution and for your insightful comments. It seems that the reference items [1-2] in the review are unfortunately missing, so this rebuttal will be based on our understanding of general related literatures. We would really appreciate it and would be happy to discuss in more details if you could provide these references during discussion period.
>
> **Distinction from previous work and new insights.**
> - All of the previous works that used both score-based model and plug-and play, e.g. [RS18, LBA+22, BB23, CDC23]  are more or less heuristic and do not have provable consistency or robustness. While some of them realized the potential of split Gibbs sampling as a plug-and-play posterior sampler, none of them found a way to solve the denoising step rigorously. One of our new insights is to show that the denoising step can be solved in full rigor with a reverse diffusion process using only unconditional score function, and it is even possible to make this diffusion process a DDIM one which enables application of many acceleration techniques. Moreover, the gain of our denoising diffusion step is not only mathematical rigor, but also improved robustness, since by running a full, multiple-step reverse diffusion process instead of a one-step heuristic denoiser as in previous works, we are able to better compensate for the drifts from the data manifold induced by the consistency step. It is with this insight that we are able to establish the first provably consistent and robust diffusion posterior sampling algorithm.
>
> **Demonstration of convergence.**
> - To further demonstrate the convergence of our algorithm as shown by our theory, we run a proof-of-concept experiment with data generated from a Gaussian mixture model, so that the convergence of distribution can be examined directly. The detailed settings can be found in General Response to All Reviewers, and the results can be found in the rebuttal pdf. It can be clearly verified from these results that our algorithm does converge to the true posterior distribution.
>
> **Additional noise in DDS-DDPM.**
> - Thank you for your sharp observations. As far as we can see, only the second row in Table 5 has noticeable noise for DPnP-DDPM. However, we also note that in this row, DDS-DDPM also recovers visibly finer details and textures of the board and the text in the original image. This tradeoff between the capability of reconstructing finer details and the risk of introducing additional noise is indeed a general phenomenon that has been observed in previous works, e.g., in [SKZ+23, page 21].
>
> **Typos.**
> - Thank you for your careful reading. We will proofread our paper more carefully and correct this typo among a few others in the final version.

---

> > ### Comment · Reviewer_ppwi · 2024-08-09
> > **Response to the author rebuttal**
> >
> > The reviewer has read the authors' rebuttal as well as the comments from other reviewers. Based on these, the reviewer prefers to maintain the initial score.

---

> > > ### Comment · Reviewer_ppwi · 2024-08-09
> > > **Further comments that may be helpful to improve the paper**
> > >
> > > The experiments presented are inadequate, and certain experimental outcomes are peculiar and fail to substantiate the claims made in the paper. For instance, the results for DPnPDDPM depicted in Table 1 are smooth, whereas those in Table 5 exhibit significant noise. The author did not address these issues during the rebuttal phase.

---

> > > > ### Author Response · Authors · 2024-08-09
> > > > **Thank you for your prompt reply, and clarification on our rebuttal**
> > > >
> > > > Thank you so much for responding to our rebuttal in a timely manner! We really appreciate it. We also appreciate your voicing of concerns regarding the experiments, and would like to clarify further on the performance DPnP-DDPM. In fact, we have addressed this under the headline of "Additional noise in DDS-DDPM" (we apologize the our algorithm name DPnP-DDPM was misspelled by DDS here due to auto-correction). We have to condense the review due to the character limit, and it is likely that you have missed it and therefore we want to repeat it below:
> > > > > - Thank you for your sharp observations. As far as we can see, only the second row in Table 5 has noticeable noise for DPnP-DDPM. However, we also note that in this row, DPnP-DDPM also recovers visibly finer details and textures of the board and the text in the original image. This tradeoff between the capability of reconstructing finer details and the risk of introducing additional noise is indeed a general phenomenon that has been observed in previous works, e.g., in [SKZ+23, page 21].
> > > >
> > > > In addition, we want to highlight that Table 1 is results for phase retrieval, while Table 5 is for quantized sensing, which are **different measurement forward models**. Therefore, their results are not directly comparable, and visually they may appear quite different.
> > > >
> > > > Again, we are happy to include new experiments to further substantiate our paper, if you are willing to provide further feedback. Thank you again for engaging with us!

---

### Official Review · Reviewer_tiqf · 2024-07-08

**Soundness:** 3
**Presentation:** 3
**Contribution:** 3
**Rating:** 6
**Confidence:** 3

**Summary:**

This paper introduces a diffusion-based sampling framework closely related to plug-and-play methods for solving general inverse problems. The technique alternates between two steps: calling a proximal consistency sampler that enforces data-fidelity, and regularization via a denoising diffusion sampler leveraging strong diffusion-based image priors.  Theoretical results demonstrate asymptotic consistency and robustness to sampling errors. Numerical experiments show promising reconstruction quality.

**Strengths:**

- The theoretical analysis is a valuable contribution. A lack of robustness to sampling errors and the resulting error accumulation has been a key challenge of diffusion-based solvers, especially in highly nonlinear tasks such as phase retrieval.

- The paper is well-written overall and the structure is logical.

- The experimental results are promising. In particular the proposed plug-and-play sampler achieves significant improvement over DPS, a well-established technique in the literature.

**Weaknesses:**

- The experimental evaluation is somewhat lacking. It would be interesting to see comparison with more contemporary solvers such as ReSample [1], which has improved robustness against sampling errors due to a posterior mean correction scheme. Moreover, in-depth ablation studies on the multiple hyperparameters of the algorithm are missing. Thus, it is unclear how much hyperparameter tuning is necessary.
- The proposed technique appears to have a very high compute cost (3000 NFEs). More discussion on the compute requirements and possible ways to accelerate the algorithm would be valuable.

[1] Song, Bowen, et al. "Solving Inverse Problems with Latent Diffusion Models via Hard Data Consistency." The Twelfth International Conference on Learning Representations.

**Questions:**

- Can the findings be extended to latent domain samplers? This would greatly improve the efficiency of the technique.
- How does the method compare to other samplers such as ReSample?
- How does performance scale with NFEs?
- I would recommend changing DDS to some other abbreviation to avoid confusion with the Decomposed Diffusion Sampling method [2].
- What does G denote in line 136?

[2] Chung, Hyungjin, Suhyeon Lee, and Jong Chul Ye. "Fast diffusion sampler for inverse problems by geometric decomposition." arXiv preprint arXiv:2303.05754 3.4 (2023).

**Limitations:**

Limitations are not clearly addressed in the paper.

---

> ### Author Rebuttal · Authors · 2024-08-04
>
> Thank you for your careful review and insightful suggestions. Below we address your concerns in a point-to-point manner.
>
> **Comparison with ReSample [SKZ+23].**
> - Thank you for your suggestion. There are a few recent algorithms that included a stochastic correction step for better robustness. The LGD-MC algorithm [SZY+23], which we compared against in our paper, is also one of them. We chose it due to its simplicity for implementation, but we are also more than happy to include comparison with the non-latent version of ReSample in the final version; comparison of latent versions, on the other hand, requires developing a latent version of our algorithm, which we feel more appropriate to leave to future work as discussed in General Response to All Reviewers. There are some preliminary comparisons in General Response to All Reviewers and in the rebuttal pdf, and we plan to include a more detailed comparison in the final version.
>
> **Computational cost.**
> - Thank you for your suggestion. In our paper, "~3000 NFEs" is the computational cost of the DDPM version of DPnP; we also proposed a DDIM version in the paper, which costs ~1500 NFEs, only 1.5x the cost of DPS (1000 NFEs). Moreover, the DDIM version allows to incorporate acceleration techniques for unconditional DDIM sampling, e.g. DPMSolver++, which can reduce the computation significantly. We would be more than happy to add more discussions in the final version.
>
> **Performance scaling with NFEs.**
> - Thank you for raising this good point. We have included a few samples with different NFEs in the rebuttal, Figure 3, and are willing to investigate it more systematically in the final version if possible.
>
> **The name DDS for Denoising Diffusion Sampler.**
> - Thank you for your suggestion. We will change it to an unambiguous name in the final version.
>
> **Notation $G$ in line 136.**
> - Thank you for your very careful reading and sharp observation. This is a typo for $M$, the forward diffusion process. We will proofread the paper carefully and make corrections in the final version.

---

### Official Review · Reviewer_43oG · 2024-07-15

**Soundness:** 3
**Presentation:** 2
**Contribution:** 2
**Rating:** 4
**Confidence:** 4

**Summary:**

This paper focuses on developing a plug-and-play algorithm for using score-based diffusion models as an expressive prior for solving nonlinear inverse problems with general forward models. While going from current state $x_{k+1}$ to $x_{k}$, this paper makes two gradient updates: (1) from $x_{k+1}$ to $x_{k+\frac{1}{2}}$ using gradients from the measurement error and (2) from $x_{k+\frac{1}{2}}$ to $x_k$ using denoising diffusion score function. Theoretically, the authors prove robustness of the proposed algorithm in sampling from the posterior, and empirically, they show that the proposed algorithm outperforms one commonly used baseline DPS.

**Strengths:**

1. This paper provides a robust algorithm for solving nonlinear inverse problems using unconditional score-based diffusion priors.
2. The paper is nicely written and theoretical analysis in a simple setting clearly demonstrates the major contributions of the paper.

**Weaknesses:**

### **Weaknesses and comments**

1. Line 171: "Assumption on forward model is applicable to *many* applications of interest" what are the applications?
2. Assumption 1: what are some examples of $\mathcal{L(\cdot, y)}$ that are differentiable almost everywhere and used in practice?
3. Step 1 in **A stochastic  DDPM-type sampler via heat flow**: How do you know the noise level $\eta$? This is typically unknown.
4. The paper is overloaded with lots of notations. To sample from the posterior, you need the measurement conditional score. By Bayes' theorem you can write this term as the combination of prior and likelihood. How do you get the unknown likelihood term exactly? Please explain the key idea without overloading with notations.
5. Inverting $\Sigma$ is difficult in high-dimension posterior sampling problems. How do you get around this issue unless you are making further approximations like diagonal covariance or scalar? In that case, how do you sample from the posterior exactly?
6. One of the interesting parts of the theoretical results in this line of research is the characterization of the discretization error. But this seems to be out of scope of this paper according to the authors (line 299).
7. Theoretical results are valid provided you get a tight TV bound for both the consistency and diffusion samplers. Any ideas how to obtain these bounds?
8. Experimental results are compared with DPS which the authors claim to be state-of-the-art. Many of the recently developed methods [1,2,3,4] outperform DPS and there is no comparison with any stronger baselines. See citations below and references therein.
9. For evaluation which dataset is used. Is it the same subset used in the original paper and other follow-up papers? The results differ quite a lot if you pick some smaller subset.
10. The authors are encouraged to cite the published version of the papers where applicable (see for instance CCL+22).


References

1. Solving Linear Inverse Problems Provably via Posterior Sampling with Latent Diffusion Models.
2. Prompt-tuning latent diffusion models for inverse problems
3. Beyond First-Order Tweedie: Solving Inverse Problems using Latent Diffusion
4. Tweedie Moment Projected Diffusions for Inverse Problems

**Questions:**

Please see the weakness section above.

**Limitations:**

Yes, the authors have addressed the limitations.

---

> ### Author Rebuttal · Authors · 2024-08-04
>
> Thank you for your detailed review and valuable comments. Below we address your comments in a point-to-point manner.
>
> **Examples for Assumption 1 on the forward model and the likelihood (point 1,2).**
> - A typical scenario is where the measurement noise $\xi$ in the measurement model $y=\mathcal{A}(x^\star)+\xi$ is Gaussian, e.g., $\xi\sim\mathcal N(0,\sigma^2I_m)$. In such case, we have$$\mathcal L(x;y)=\log p(y|x^\star=x)=-\frac1{2\sigma^2}\Vert y-\mathcal A(x)\Vert^2 - \frac{m}2\log(2\pi\sigma^2).$$
> Therefore Assumption 1 holds for *all* applications where the forward model $\mathcal A$ is almost everywhere differentiable, in particular, if $\mathcal A$ is linear. This includes inpainting, deblurring, super-resolution, phase retrieval, etc. It is noteworthy that most of the existing works (e.g. [CKM+23, SKZ+23, SZY+23, DS24, MSKV24], and [1-4] you listed), have the same differentiability assumption or requires the stronger condition that $\mathcal A$ is linear.
>
> **The noise level $\eta$ (point 3).**
> - The noise level $\eta$ in the DPnP framework is a hyperparameter used for annealing. It is not related to, and should not be confused with, the *measurement noise level* (the variance of $\xi$ in the measurement model $y=\mathcal A(x^\star)+\xi$). The annealing noise level at the $k$-th iteration, denoted by $\eta_k$ in Algorithm 1, is set manually, hence is always known.
>
> **How to know the likelihood term (point 4).**
> - As the example above (please refer to our response to point 1 and 2) demonstrates, the likelihood can be computed if the forward model $\mathcal{A}$ and the noise distribution is known. This setting is adopted by most of the existing works (please refer to our response to point 1,2), including [1-4] you listed (in fact, most of them assumed $\mathcal{L}(x;y)=-\frac1{2\sigma^2}\Vert y-\mathcal A(x)\Vert^2 + \rm const$, which is a special case of ours with Gaussian measurement noise). It might also be helpful to note that our log-likelihood function $\mathcal L$ is fully, simply determined by the measurement model, and should not be confused with $\log p(y|x_t)$, which was also called likelihood by some of the previous works.
>
> **Computation cost of inverting $\Sigma$ (point 5).**
> - While most of the existing works (including [1-4]) simply assumed (implicitly or explicitly) $\Sigma=\frac1{\sigma^2}I$ is scalar (please refer to our response to point 4), it is possible to handle general $\Sigma$ with standard numerical tricks in our work. For example, one can pre-compute, *once and for all*, the vector $A^\top\Sigma^{-1}y$ (which amounts to solving a linear equation) and a prefactorizization of the PSD matrix $A^\top\Sigma^{-1}A$. Using such precomputed information, each iteration only needs to compute several matrix-vector products, which has a negligible computation cost compared to Neural Function Evaluations. The cost of computing the prefactorization of $A^\top\Sigma^{-1} A$ has also been observed to be acceptable [SVMK22, KEES22]. Another option is to use MALA for the PCS step as we did for general non-linear inverse problems, so that there is no need to invert any matrix.
>
> **Discretization error and TV bound for subsamplers (point 6,7).**
> - As Theorem 2 indicates, discretization error affects our algorithm only through total variation error of subsamplers DDS and PCS, so we respond to point 6 and 7 together. The reason we did not include an analysis of TV error of subsamplers is, as explained in the paper, that both subsamplers can be analyzed with existing techniques in the references we provided. For example, applying the results in [LWCC23], one may show that DDS-DDPM has total variation error $$\varepsilon_{\sf DDS-DDPM}\le \tilde C\frac{d^2}{\sqrt{T'}}+\tilde C\sqrt d\varepsilon_{\sf score},$$ where $\tilde C$ hides logarithmic dependence on parameters, $T'$ is the number of steps (cf. Algorithm 2), $\varepsilon_{\sf score}$ is score estimation error defined in [LWCC23]. Similar bound for DDS-DDIM can also be proved using the results therein. On the other hand, utilizing results on the convergence of MALA [CLA+21], one may prove that PCS has total variation error $$\varepsilon_{\sf PCS}\le C\exp(-C\sqrt{N}\eta^{-1/6}d^{-1/4}).$$
>
> **Comparison with works other than DPS (point 8).**
> - Thank you for the references. As we mentioned in General Response to All Reviewers, we compare our algorithm not only with DPS, but **also with more recent, highly competitive** algorithm like LGD-MC [SZY+23], and plan to add comparison with ReSample (pre version available in the rebuttal pdf). Note that comparison with such stronger baseline as we have done is actually not too common in the literature, given the highly active status of the field and the well-established position of DPS, e.g. in all the above references and [1-4] you listed, the strongest non-latent baseline is DPS (latent models are discussed in the next paragraph). In addition, none of these previous works come with a provable robustness guarantee, thus our algorithm is also of independent theoretical interest.
> - Concerning [1-4], all of them except [2] are confined to linear inverse problems, while our focus is on general, non-linear inverse problems (e.g. phase retrieval and quantized sensing). [1-3] rely on the use of latent diffusion models ([2] further requires prompting the model) which is again a narrower setting than ours as discussed in General Response to All Reviewers. Unavailability of code also hinders comparison with [2-4].
>
> **Whether the dataset is the same with DPS (point 9).**
> - We have strictly followed the original DPS paper on the usage of dataset. We used the same dataset without picking smaller subset than DPS. Our experiments are done for different inverse problems, though, as our focus is on more non-linear problems. In doing so, we made our best effort to ensure fair comparison by fine-tuning all competing algorithms and listing explicitly our problem parameters in the paper.

---

> ### Author Response · Authors · 2024-08-12
> **Thank you for your review. Have we addressed your concerns?**
>
> Dear Reviewer 43oG,
>
> We've taken your initial feedback into careful consideration in our response. Could you kindly confirm whether our responses have appropriately addressed your concerns?
>
> If you find that we have properly addressed your concerns, could you please kindly consider increasing your initial score accordingly? Please let us know if you have further comments.
>
> Thank you for your time and effort in reviewing our work!
>
> Many thanks, Authors

---

> > ### Comment · Reviewer_43oG · 2024-08-12
> > **Discussion with Authors**
> >
> > The reviewer thanks the authors for the detailed response. The reviewer is satisfied with the clarifications in Q1, Q2, Q3 and Q9. However, the major concerns still remain. Regarding Q4, the term $\log p(y|x_t)$ is typically not computed and existing methods (e.g. DPS) approximate this using $\log p(y|E[x_0|x_t])$, which is similar to the term $L(x;y)$ used in this paper except the additive score function and a cross-term from the Tweedie's formula. Since the noise level $\sigma$ is anyway not known and needs to be tuned in stepsize, the reviewer suspects that the proposed method doesn't offer any major advantages over existing methods at the cost of more compute. This observation is also supported by experiments in Tables 2 and 3.
> >
> > For Q5, how would you precompute $A^T\Sigma^{-1}y$ for inverse problems typically considered in practice, such as Gaussian deblur, motion deblur or super-resolution? What would be the storage space complexity in high-dimensional applications where images could be of size 1024x1024? The reviewer thinks that these issues have not been properly addressed in the paper.
> >
> > For Q6 and Q7, there is no discussion regarding this important piece of information in the main paper, which would essentially help the reader when to choose this algorithm over others. Especially, the dependence on $d$ in $\varepsilon_{DDS-DDPM}$ seems problematic for large-scale applications with $d$ of the order $10^6$ as in recent state-of-the-art inverse solvers.
> >
> > For Q8, the compared baselines are weak and do not adequately justify the claims of the paper. The reviewer was referring to more recently developed pixel space diffusion based inverse solvers such as TMPD or ReSample. The reviewer thanks the authors for providing some preliminary experiments on ReSample.
> >
> > The reviewer will follow the guidelines in revising the score if needed after all the questions have been addressed properly.

---

> > > ### Author Response · Authors · 2024-08-12
> > > **Thank you for your response, and further clarification**
> > >
> > > Thank you for engaging with us! We are happy to hear that many of your concerns have been addressed successfully, and appreciate your detailed comments that provide us an opportunity to clarify further the remaining points.
> > >
> > > Regarding your comments on Q4, we would like to clarify two points:
> > > - Our approach does not involve approximating $\log p(y|x_t)$ as in the previous algorithms. Our $\mathcal L(x; y)$ is not an approximation to $\log p(y|x_t)$; it is simply the likelihood $\log p(y|x_0)$ (where $x_0$ is the ground truth signal), using the notation in DPS paper, which is assumed known in most of the previous works. Our approach deviates significantly from DPS; the reason we were able to bypass this approximation is that we directly tackle the whole posterior distribution $\propto p^\star(x_0)p(y|x_0) = p^\star(x_0) \exp(\mathcal{L}(x_0; y))$ using a split Gibbs sampler, which is made practical by developing proximal samplers for both $p^\star(x_0)$ and $\exp(\mathcal{L}(x_0; y))$ with diffusion and MALA.
> > > - The measurement noise level $\sigma$ is assumed known or is a tunable parameter, which is consistent with most of the previous works (e.g. DDRM, DPS, LGD-MC, ReSample). As can be seen from the experimental results (including our rebuttal pdf), our algorithm demonstrates significant improvement for highly non-linear problems like phase retrieval, over recent works like LGD-MC and ReSample.
> > >
> > > Regarding your comments on Q5, we would like to note that (i) it is a common choice as in many previous works (e.g. the references above) that $\Sigma$ is chosen as a scalar identity, in which case it is not necessary to perform a matrix inversion; (ii) The storage cost of prefactorization has been found to be managable (of $O(n)$ order) with memory efficient SVD in many practical inverse problems, e.g. denoising, inpainting, super resolution, deblurring, and colorization, cf. DDRM [KEES22]; (iii) If memory is really of concern, there is also the option to simply use MALA for the proximal consistency step that avoids direct inversion, which is still theoretically sound as our theory tolerates errors for both subsamplers.
> > >
> > > Regarding your comments on Q6 and Q7, we would be happy to include more discussion in the final paper when space permits, which we suppressed in the submission and left a few references. Note that the dependence on $d$ in $\varepsilon_{\sf DDS-DDPM}$ can be further improved to $\tilde{O}(\sqrt{d/T}) + O(\varepsilon_{\sf score})$ using sharper results in [BDBDD24]. However, such dependency with $d$ is known to be tight and generally non-avoidable when plain diffusion models are used.
> > >
> > > Regarding your comments on Q8, since there is only one day left before the discussion deadline, it is challenging to provide more experiments result in time before the discussion period ends. Nonetheless, we are committed to include more algorithm evaluation in the final version. We also want to provide a bit more discussion regarding the additional experimental results regarding the full evaluation of ReSample on FFHQ dataset, which is already in the rebuttal pdf. Therein, it can be seen that LGD-MC, one of the baseline in our original submission, is a competitive baseline with performance close to that of ReSample. Our algorithm demonstrates significant advantages over both, especially on the phase retrieval task. We expect similar conclusions will hold when we compare ReSample on other datasets/tasks.
> > >
> > > Thank you again for your careful review. We appreciate your constructive feedback and are happy to discuss more.

---

### Author Rebuttal · Authors · 2024-08-04

# General Response to All Reviewers

We would like to express our cordial thanks to all the reviewers for their careful review and constructive feedback. Below we address some common concerns raised by the reviewers. Our point-to-point response can be found in the separate rebuttal to each reviewer.

**Comparison with latent diffusion.** Our work is complementary (rather than parallel) to previous literatures on latent diffusion. It is mostly straightforward to incorporate latent diffusion models into our framework (it suffices to modify the Denoising Diffusion Sampler in our framework to a latent version), which will hopefully further allow our algorithm to take advantage of latent diffusion models. On the other hand, such pre-trained latent model is not always available for imaging tasks (e.g. the experiments of our work and many previous works [CKM+23, CLY23, MSKV24]), and our framework still works when only non-latent models are available. We opt to present only the non-latent version in the paper to better highlight the key novel ideas of our work (together with its theory) within limited pages and without compromising generality of settings, which is already quite lengthy. However, we agree that incorporating latent diffusion is a promising direction for future research.

**Experimental evaluation.** We compare our algorithm with the well-established DPS algorithm **as well as more recent, highly competitive algorithm** such as LGD-MC [SZY+23] which already demonstrates significant advantage over DPS. We abide to the experimental settings in these previous works with best efforts, and the size of testing dataset is kept strictly the same as that in DPS. Following the suggestions of reviewers, **we have also included a brief comparison with ReSample [SKZ+23] below and in the rebuttal pdf**, which we will try to make more complete in the final version. In addition, we would like to emphasize that none of the previous works have a provable robustness guarantee, thus our work is also of independent theoretical interest.

------------------------------
# Rebuttal pdf
Please find in attached our one-page pdf containing the additional experimental results cited in our responses. The settings of these results are explained below.

**Comparison with ReSample.** For reasons explained above, we compare our algorithm with the non-latent (also termed "pixel-based" in the [SKZ+23]) version of ReSample, and leave the comparison of latent versions for future work after we developed a latent version of our algorithm. For ReSample, we use DDIM with $T=1000$ steps, which is the same as all the other algorithms. We let ReSample run without stochastic resampling for the first $250$ steps as specified in the original paper. After that, we run stochastic resampling every $10$ steps as in the original paper. All other parameters, including the resampling noise factor $\gamma$, the (conjugate) gradient stepsize, early stopping criterion, etc., has been fine-tuned with reasonable effort for best performance. We evaluate its performance on the same set of inverse problems on FFHQ-1k dataset as in our paper, i.e. super-resolution, phase retrieval, and quantized sensing (problem parameters was given in our paper). **The results are shown in Table 1 in the rebuttal pdf.** We make a few comments on the results here.
- Non-latent ReSample does not work on phase retrieval, despite our considerable effort in trying different parameters. This may be accounted by the complicated non-linear, non-convex landscape of the phase retrieval objective, as in such case the hard consistency enforcing step in ReSample may easily stuck at local minimum. It is possible that using the latent version of ReSample may alleviate this issue, but as we said, this awaits future research.
- Overall, on the tasks where ReSample works, i.e., super-resolution and quantized sensing, the performance of ReSample is better than DPS and close to that of LGD-MC which we already compared with. All of DPS, LGD-MC and ReSample are surpassed by our algorithm in quantitative metrics.

**Demonstration of convergence.** We verify our theoretical claim of convergence of our algorithm on a Gaussian mixture model as a toy example. We consider the setting where the unconditional distribution $p^\star(x)$ is a 2D GMM given by
$$x = [x_1, x_2] \sim 0.6 \cdot\mathcal N([-3, -1]^\top, I_2) + 0.4 \cdot\mathcal N([1, 1]^\top, I_2).$$
We consider a linear inverse problem where $\mathcal{A}(x) = x_1-x_2$, and suppose the measurement $y=-0.5$. For simplicity of demonstration, we further consider the noiseless setting, so that the posterior distribution $p(x|y)$ is a degenerate distribution supported on the 1D line $\\{x\in\mathbb{R}^2: y=\mathcal{A}(x)\\}=\\{x\in\mathbb{R}^2: x_1-x_2=-0.5\\}$. **A depiction of the setting is given in Figure 2.(a) in the rebuttal pdf.** We run DPnP on this linear inverse problem assuming the unconditional score function is exactly known (which can be computed from the GMM expression). We use an annealing schedule with $K=120$, $\eta_0=0.5$, $\eta_K=10^{-4}$. **The distribution of $\hat x_k$, the $k$-th iterate of DPnP, is shown in Figure 1 in the rebuttal pdf.** We further compute the total variation distance between the distribution of $\hat x_k$ and the true posterior distribution. **The result of total variation error is shown in Figure 2.(b) in the rebuttal pdf.** It can be inferred clearly that the distribution of the iterates of DPnP eventually converges (in total variation) to the true posterior distribution.

---

### Decision · Program_Chairs · 2024-09-25

**Decision:**

Accept (poster)

**Comment:**

In this paper, the authors present a plug-and-play algorithm based on score-based diffusion models as a signal prior for a variety of linear and nonlinear inverse problems.  In contrast to existing diffusion based approaches for inverse problems, the authors prove a robustness guarantee for their proposed method.  The proposed method also demonstrates superior performance in a variety of metrics on certain inverse problems.  The strengths of this paper are the novel theoretical guarantee of robustness in a diffusion based algorithm, and the flexibility of the proposed algorithm to handle a variety of diffusion models.  A weakness of the paper is that the experimental investigations could be more thorough.  Overall, the strengths outweigh the weaknesses, and the paper is recommended for acceptance.